# Towards Human-AI Complementarity
# with Prediction Sets

**Giovanni De Toni**[*]
Fondazione Bruno Kessler & University of Trento
Trento, Italy
giovanni.detoni@unitn.it

**Nastaran Okati**
Max Planck Institute for Software Systems
Kaiserslautern, Germany
nastaran@mpi-sws.org

**Suhas Thejaswi**
Max Planck Institute for Software Systems
Kaiserslautern, Germany
thejaswi@mpi-sws.org

**Eleni Straitouri**
Max Planck Institute for Software Systems
Kaiserslautern, Germany
estraitouri@mpi-sws.org

**Manuel Gomez-Rodriguez**
Max Planck Institute for Software Systems
Kaiserslautern, Germany
manuelgr@mpi-sws.org

## Abstract

Decision support systems based on prediction sets have proven to be effective at helping human experts solve classification tasks. Rather than providing single-label predictions, these systems provide sets of label predictions constructed using conformal prediction, namely prediction sets, and ask human experts to predict label values from these sets. In this paper, we first show that the prediction sets constructed using conformal prediction are, in general, suboptimal in terms of average accuracy. Then, we show that the problem of finding the optimal prediction sets under which the human experts achieve the highest average accuracy is NP-hard. More strongly, unless $P = NP$, we show that the problem is hard to approximate to any factor less than the size of the label set. However, we introduce a simple and efficient greedy algorithm that, for a large class of expert models and non-conformity scores, is guaranteed to find prediction sets that provably offer equal or greater performance than those constructed using conformal prediction. Further, using a simulation study with both synthetic and real expert predictions, we demonstrate that, in practice, our greedy algorithm finds near-optimal prediction sets offering greater performance than conformal prediction.

## 1 Introduction

In recent years, there has been increasing excitement about the potential of decision support systems based on machine learning to help human experts make more accurate predictions in a variety of application domains, including medicine, education and science [1–3]. In this context, the ultimate goal is human-AI complementarity—the predictions made by the human expert who uses a decision support system are more accurate than the predictions made by the expert on their own and by the classifier used by the decision support system [4–8].

The conventional wisdom is that to achieve human-AI complementarity, decision support systems should help humans understand when and how to use their predictions to update their own. As a result,

---

[*]The author contributed to this paper during an internship at the Max Planck Institute for Software Systems.

38th Conference on Neural Information Processing Systems (NeurIPS 2024).

a flurry of empirical studies has analyzed how factors such as confidence, explanations, or calibration influence when and how humans use the predictions provided by a decision support system [9–12]. Unfortunately, these studies have been so far inconclusive and it is yet unclear how to design decision support systems that achieve human-AI complementarity [13–17].

In this context, Straitouri et al. [18, 19] have recently argued, both theoretically and empirically, that an alternative type of decision support systems may achieve human-AI complementarity, by design. Rather than providing a single label prediction and letting a human expert decide when and how to use the predicted label to update their own prediction, these systems provide a set of label predictions, namely a prediction set, and ask the expert to predict a label value from the set.[2] To construct each prediction set, these systems rely on a conformal predictor [20, 21]. The conformal predictor first computes a non-conformity score for each potential label value using the output provided by a classifier (*e.g.*, the softmax scores), and then adds a label value to the prediction set if its non-conformity score is below a data-driven threshold computed using a calibration set. Further, to optimize the performance of these systems, Straitouri et al. have introduced several methods to efficiently find the optimal value of the threshold used by the conformal predictor.[3] However, it is unclear whether the optimal prediction sets maximizing the average accuracy achieved by an expert who uses such systems can always be constructed using a deterministic threshold rule as the one used by a conformal predictor. Motivated by this observation, in this work, our goal is to understand how to construct optimal prediction sets under which human experts achieve the highest average accuracy.

**Our contributions.** We first demonstrate that there exist (many) data distributions for which the optimal prediction sets under which the human experts achieve the highest average accuracy cannot be constructed using a conformal predictor. Then, we show that the problem of finding the optimal prediction sets is NP-hard by using a reduction from the $k$-clique problem [22]. More strongly, unless P = NP, we show that the problem is hard to approximate to any factor less than the size of the label set. However, we introduce a simple and computationally efficient greedy algorithm that, for a large class of non-conformity scores and expert models parameterized by a mixture of multinomial logit models (MNLs), is guaranteed to find prediction sets that provably offer equal or greater performance than those constructed using conformal prediction. Moreover, using a simulation study with both synthetic and real expert predictions, we demonstrate that, in practice, our greedy algorithm finds near-optimal prediction sets offering greater performance than conformal prediction. We have released an open-source implementation of our greedy algorithm as well as the code and data used in our experiments at `https://github.com/Networks-Learning/towards-human-ai-complementarity-predictions-sets`.

**Further related work.** Our work builds upon further related work on set-valued predictors, assortment optimization, and learning under algorithmic triage.

The literature on set-valued predictors aims to develop predictors that, for each sample, output a set of label values, namely a prediction set [23]. Set-valued predictors have not been designed nor evaluated by their ability to help human experts make more accurate predictions [24–27], except for a few notable exceptions [18, 19, 28–30]. These exceptions provide empirical evidence that conformal predictors, a specific type of set-valued predictors, may help human experts make more accurate predictions. Among these exceptions, the work by Straitouri et al. [18, 19], which we have already discussed previously, is most related to ours. In this context, it is also worth noting that a recent theoretical study has argued that prediction sets may also help experts create more accurate rankings [31].

The literature on assortment optimization aims to develop methods to help a seller select a subset of products from a universe of substitutable products, namely an assortment, with maximum expected revenue [32–36]. Within this literature, the work most closely related to ours tackles the assortment optimization problem under customization [35, 36], where there are different types of customers and each type of customer chooses products following a different multinomial logit model. More specifically, by mapping products to label values, types of customers to ground truth label values, and revenue to accuracy, one could think of our problem as an assortment optimization problem

---

[2]There are many decision support systems used by experts that, under normal operation, forcefully limit experts' level of agency. For example, in aviation, there are automated, adaptive systems that prevent pilots from taking certain actions based on the monitoring of the environment.

[3]A threshold value is optimal if it maximizes the average accuracy achieved by an expert who predicts label values from the prediction sets created by the conformal predictor.

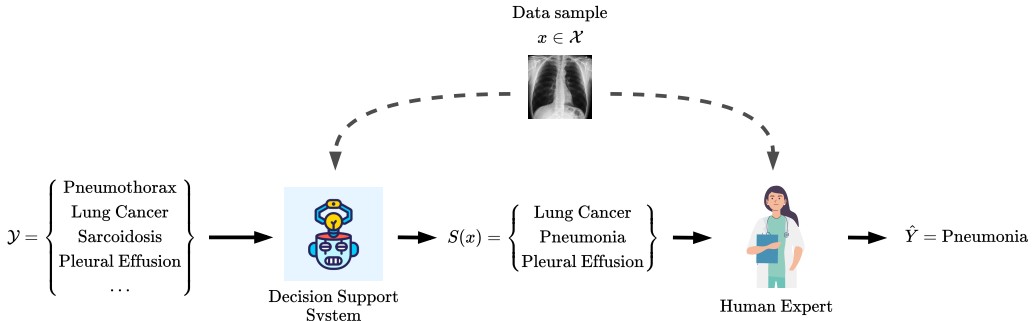

Figure 1: Our automated decision support system. Given an instance with a feature vector $x$, the system $\mathcal{C}$ helps the expert by automatically narrowing down the set of potential label values to a prediction set $\mathcal{S}(x) \subseteq \mathcal{Y}$. The system asks the expert to predict a label value $\hat{y}$ from $\mathcal{S}(x)$.

under customization. However, in the assortment optimization problem under customization, the type of each customer is known and thus may be offered different subsets of products whereas, in our problem, the ground truth label is unknown. As a result, (the complexity of) our problem and our technical contributions are fundamentally different.

The literature on learning under algorithmic triage aims to develop classifiers that make predictions for a given fraction of the samples and leave the remaining ones to human experts, as instructed by a triage policy [37–42]. In contrast, in our work, for each sample, a classifier is used to construct a prediction set and a human expert needs to predict a label value from the set. In this context, it is also worth noting that learning under algorithmic triage has been extended to reinforcement learning settings [43–46].

## 2 Decision Support Systems Based on Prediction Sets

Given a multiclass classification task where, for each task instance, a human expert needs to predict the value of a ground truth label $y \in \mathcal{Y} = \{1, \ldots, L\}$, we focus on the design of a decision support system that, given a set of features $x \in \mathcal{X}$, helps the expert by narrowing down the set of potential label values to a subset of them $\mathcal{S}(x) \subseteq \mathcal{Y}$. Here, similarly as in Straitouri et al. [18, 19], we assume that, for any instance with features $x \in \mathcal{X}$, the system asks the expert's prediction $\hat{y} \in \mathcal{Y}$ to belong to the prediction set $\mathcal{S}(x)$, *i.e.*, $\hat{y} \in \mathcal{S}(x)$. The key rationale for restricting the expert's agency is that, if we would allow the expert to predict label values from outside the prediction set, a good performance would depend on the expert developing a good understanding of when to predict a label from the prediction set. In this context, it is worth highlighting that Straitouri et al. [19] run a large-scale human subject study to compare the above setting against an alternative setting where experts are allowed to predict label values from outside the prediction sets. They found that, in the alternative setting, the number of predictions in which the prediction sets do not contain the true label and the experts succeed is consistently smaller than the number of predictions in which the prediction sets contain the true label and the experts fail. As a consequence, in the alternative setting, experts perform worse. Refer to Figure 1 for an illustration of the decision support system.

Then, for any $x \in \mathcal{X}$, our goal is to find the optimal prediction set $\mathcal{S}^*(x)$ that maximizes the average accuracy of the expert's prediction,[4] *i.e.*,

$$\mathcal{S}^*(x) = \underset{\mathcal{S} \subseteq \mathcal{Y}}{\operatorname{argmax}} \, g(\mathcal{S} \,|\, x) \quad \text{where} \quad g(\mathcal{S} \,|\, x) = \mathbb{E}_{Y \sim P(Y \,|\, X), \hat{Y} \sim P_{\mathcal{S}}(\hat{Y} \,|\, X, Y)} \Big[ \mathbb{I}\{\hat{Y} = Y\} \,|\, X = x \Big],$$
(1)

where $P(Y \,|\, X)$ denotes the conditional distribution of the ground-truth label $Y$ and $P_{\mathcal{S}}(\hat{Y} \,|\, X, Y)$ denotes the conditional distribution of the expert's predictions $\hat{Y}$ under the prediction set $\mathcal{S}$.[5]

---

[4]We denote random variables with capital letters and realizations of random variables with lowercase letters.

[5]The expert's prediction $\hat{Y}$ and the ground truth label $Y$ may *not* be conditionally independent given the set of features $X$ since, in most application domains of interest, the expert may have access to additional features. Otherwise, one may argue that pursuing human-AI complementarity is not a worthy goal [47].

# 3 On the Suboptimality of Conformal Prediction

Given a user-specified parameter $\alpha \in [0, 1]$, a conformal predictor uses a choice of non-conformity score $s : \mathcal{X} \times \mathcal{Y} \to \mathbb{R}$ and a calibration set $\mathcal{D}_{\text{cal}} = \{(x_i, y_i)\}_{i=1}^m$, where $(x_i, y_i) \sim P(X)P(Y \mid X)$, to construct the prediction sets $\mathcal{S}(X) = \mathcal{S}_{\text{cp}}(X)$ as follows:

$$\mathcal{S}_{\text{cp}}(X) = \{y \mid s(X, y) \leq \hat{q}_\alpha\}, \tag{2}$$

where $\hat{q}_\alpha$ is the $\lceil (m + 1)(1 - \alpha) \rceil / m$ empirical quantile of the non-conformity scores of the samples in the calibration set $\mathcal{D}_{\text{cal}}$. By using the above construction, the conformal predictor guarantees that the probability that the true label $Y$ belongs to the subset $\mathcal{S}_{\text{cp}}(X)$ is almost exactly $1 - \alpha$, i.e., $1 - \alpha \leq P(Y \in \mathcal{S}_{\text{cp}}(X)) \leq 1 - \alpha + 1/(m + 1)$, as shown elsewhere [20, 21].

Under common choices of non-conformity scores [48, 49], there are many data distributions for which the optimal prediction set under which the human expert achieves the highest accuracy cannot be constructed using a conformal predictor. Consider the following example where $\mathcal{Y} = \{1, 2, 3\}$ and,

$$P(Y = y \mid X = x) = \begin{cases} 0.4 & \text{if } y = 1 \\ 0.35 & \text{if } y = 2 \\ 0.25 & \text{if } y = 3 \end{cases} \text{ and } P_{\mathcal{S}}(\hat{Y} = \hat{y} \mid X = x, Y = y) = \frac{C_{\hat{y}y}}{\sum_{y' \in \mathcal{S}} C_{y'y}},$$

where $C_{1,1} = C_{2,1} = C_{3,1} = 0.33$, $C_{1,2} = C_{1,3} = 0.4$, $C_{2,2} = C_{3,3} = 0.6$, and $C_{3,2} = C_{2,3} = 0$. A brute force search reveals that the optimal prediction set is $\{2, 3\}$ and, under this set, the expert achieves accuracy 0.6. Now, assume we have access to a perfectly calibrated classifier $f(x) \in [0, 1]^L$, i.e., for all $x \in \mathcal{X}$ and $y \in \mathcal{Y}$, it holds that $f_y(x) = P(Y = y \mid X = x)$. Then, for any choice of $\alpha \in [0, 1]$, as long as the non-conformity scores rank the label set in decreasing order of $f_y(x)$, the prediction set provided by conformal prediction can only be among the sets $\{\emptyset, \{1\}, \{1, 2\}, \{1, 2, 3\}\}$. Among these sets, the set under which the expert achieves the highest accuracy is $\{1, 2, 3\}$ and, under this set, the expert achieves accuracy $0.49 < 0.6$.

Motivated by the above example, one may think of closing the above performance gap by incorporating information about the distribution of experts' predictions in the definition of the non-conformity score. However, we cannot expect to fully close the performance gap since, as we will show next, the problem of finding the optimal prediction sets is NP-hard to solve and approximate to any factor less than the size of the label set $\mathcal{Y}$.

# 4 On the Hardness of Finding the Optimal Prediction Sets

In this section, we first show that, given $x \in \mathcal{X}$, we cannot expect to find the optimal prediction set $\mathcal{S}^*(x)$ that maximizes the accuracy of the expert's prediction in polynomial time:[6]

**Theorem 1** *The problem of finding the optimal prediction set, as defined in Eq. 1, is* NP-*hard.*

In the proof of the above theorem, we first reduce the $k$-clique problem,[7] which is known to be NP-complete [22], to an instance of the problem of deciding whether there exists a prediction set $\mathcal{S} \subseteq \mathcal{Y}$ such that $g(\mathcal{S} \mid x) \geq B$ given a constant $B > 0$. More specifically, given a $k$-clique problem defined over a graph $\mathcal{G} = (\mathcal{V}, \mathcal{E})$ with $k \leq |\mathcal{V}|$, we reduce it to an instance of the above decision problem in which $\mathcal{Y} = \mathcal{V}$, $B = \frac{k}{|\mathcal{V}|}$ and, for all $y \in \mathcal{Y}$, we have that $P(Y = y \mid X = x) = \frac{1}{|\mathcal{V}|}$ and

$$P_{\mathcal{S}}(\hat{Y} = \hat{y} \mid X = x, Y = y) = \frac{C_{\hat{y}y}}{\sum_{y' \in \mathcal{S}} C_{y'y}} \text{ where } C_{y'y} = \begin{cases} 0 & \text{if } (y', y) \in \mathcal{E} \\ 1/\widehat{N}_{\mathcal{G}}(y) & \text{otherwise,} \end{cases} \tag{3}$$

and $\widehat{N}_{\mathcal{G}}(y)$ denotes the number of vertices that are not adjacent to $y$. Then, since the above decision problem can be trivially reduced to the problem of finding the optimal prediction set (in polynomial time), we conclude that the problem is NP-hard.

Motivated by the above result, we may think in looking for desirable properties for the objective function $g(\mathcal{S} \mid x)$ such as monotonicity and submodularity,[8] which would allow for the design of

---

[6]All proofs are in Appendix A.

[7]Given a graph $\mathcal{G} = (\mathcal{V}, \mathcal{E})$ and an integer $k \leq |\mathcal{V}|$, the $k$-clique problem seeks to decide whether there exists $\mathcal{S} \subseteq V$ with size $|\mathcal{S}| = k$ such that, for every $u, v \in S$, there exists $(u, v) \in \mathcal{E}$.

[8]A function $f : 2^{\mathcal{Y}} \to \mathbb{R}$ is submodular if and only if, for every $\mathcal{S} \subseteq \mathcal{T} \subseteq \mathcal{Y}$ and $y \in \mathcal{Y} \backslash \mathcal{T}$, it holds that $f(\mathcal{S} \cup \{y\}) - f(\mathcal{S}) \geq f(\mathcal{T} \cup \{y\}) - f(\mathcal{T})$.

**Algorithm 1:** Greedy algorithm

**Input:** Label set $\mathcal{Y}$, features $x$, classifier $f$, confusion matrix $\boldsymbol{C}$
**Output:** Prediction set $\mathcal{S}$

```
1  S ← ∅
2  {y_(1), ..., y_(L)} ← argsort f(x)                  // Sort in descending order
3  for k ∈ {1, ..., L} do
4  |   S_k ← ∅
5  |   while |S_k| < k do                  // Add labels to the prediction set until we hit k
6  |   |   Δ* ← -∞
7  |   |   for y ∈ {y_(1), ..., y_(k)}\S_k do
8  |   |   |   Δ ← ĝ(S_k ∪ {y} | x) - ĝ(S_k | x)   // Eval the marginal gain of adding y to S_k
9  |   |   |   if Δ > Δ* then
10 |   |   |   |   Δ* ← Δ, y* ← y
11 |   |   S_k ← S_k ∪ {y*}                  // Add label offering the largest marginal gain
12 |   |   if ĝ(S_k | x) > ĝ(S | x) then
13 |   |   |   S ← S_k                       // Update S if S_k achieves higher objective value
14 return S
```

approximation algorithms with non-trivial approximation guarantees [50]. Unfortunately, there are many data distributions for which the objective function is neither monotone nor submodular. For example, assume $\mathcal{Y} = \{1, 2, 3\}$,

$$P(Y = y \mid X = x) = \begin{cases} 0.4 & \text{if } y = 1 \\ 0.35 & \text{if } y = 2 \\ 0.25 & \text{if } y = 3 \end{cases} \text{ and } P_{\mathcal{S}}(\hat{Y} = \hat{y} \mid X = x, Y = y) = \frac{C_{\hat{y}y}}{\sum_{y' \in \mathcal{S}} C_{y'y}},$$

where $C_{1,1} = 0.2$, $C_{1,2} = C_{2,1} = C_{1,3} = C_{3,1} = 0.4$, $C_{2,2} = C_{3,3} = 0.6$ and $C_{2,3} = C_{3,2} = 0$. For $\mathcal{S} = \{1\} \subseteq \mathcal{T} = \{1, 2\} \subset \mathcal{Y}$, it holds that $g(\mathcal{S} \mid x) = 0.4 > g(\mathcal{T} \mid x) = 0.34$ and $g(\mathcal{T} \mid x) = 0.34 < g(\mathcal{Y} \mid x) = 0.44$, and thus we can conclude it is not monotone. Moreover, it also holds that $g(\mathcal{S} \cup \{3\} \mid x) - g(\mathcal{S} \mid x) = -0.116 < g(\mathcal{T} \cup \{3\} \mid x) - g(\mathcal{T} \mid x) = 0.096$, and thus we can conclude it is not submodular.

In fact, the following theorem shows that we cannot expect to find a polynomial-time algorithm to find a non-trivial approximation to our problem:

**Theorem 2** *The problem of finding the optimal prediction set, as defined in Eq. 1, is* NP-*hard to approximate to any factor less than the size $L$ of the label set $\mathcal{Y}$.*

In the proof of the above theorem, we first show that, given a polynomial-time $\alpha$-approximation algorithm for the problem of finding the optimal prediction set, we can obtain a polynomial-time $\alpha$-approximation algorithm for the problem of finding the maximum clique in a graph $\mathcal{G} = (\mathcal{V}, \mathcal{E})$.[9] Then, since it is known that, for any $\epsilon > 0$, the latter problem is NP-hard to approximate to a factor $|\mathcal{V}|^{1-\epsilon}$ [51], we can conclude that the problem of finding the optimal prediction set is NP-hard to approximate to a factor $|\mathcal{Y}|^{1-\epsilon}$.

While the above hardness results may be discouraging, in what follows, we will introduce a simple greedy algorithm that provably offers equal or greater performance than conformal prediction for a large class of non-conformity scores and expert models, and in practice, often succeeds at finding (near-)optimal prediction sets.

**A simple greedy algorithm.** Given a sample with features $x \in \mathcal{X}$ and a prediction set $\mathcal{S} \subseteq \mathcal{Y}$, our greedy algorithm estimates the accuracy of the expert's prediction, as defined in Eq. 1, using the following estimator:

$$\hat{g}(\mathcal{S} \mid x) = \sum_{y \in \mathcal{S}} \underbrace{f_y(x)}_{(a)} \underbrace{\frac{C_{yy}}{\sum_{y' \in \mathcal{S}} C_{y'y}}}_{(b)} \tag{4}$$

---

[9]Given a graph $\mathcal{G} = (\mathcal{V}, \mathcal{E})$, the maximum clique problem seeks to find the largest $\mathcal{S} \subseteq \mathcal{V}$ such that, for every $u, v \in \mathcal{S}$, there exists $(u, v) \in \mathcal{E}$.

where (a) approximates $P(Y = y \mid X = x)$ using a well-calibrated classifier $f(x) \in [0, 1]^L$ and, similarly as in Straitouri et al. [18], (b) approximates $P_\mathcal{S}(\hat{Y} = y \mid X = x, Y = y)$ using a mixture of multinomial logit models (MNLs) parameterized by the confusion matrix of the predictions made by the expert on their own, *i.e.*, $C_{y'y} = P_\mathcal{Y}(\hat{Y} = y' \mid Y = y)$.

The greedy algorithm first ranks each label value $y \in \mathcal{Y}$ using the output $f_y(x)$ of the classifier. Let $y_{(1)}, \ldots, y_{(L)}$ be the label values ordered according to such a ranking, where $\cdot_{(i)}$ denotes the $i$-th label value in the ranking and $f_{y_{(i)}}(x) \geq f_{y_{(j)}}(x)$ for all $i < j$. Then, it runs $L$ rounds and, at each $k$-th round, it starts from the prediction set $\mathcal{S}_k = \emptyset$ and iteratively adds to $\mathcal{S}_k$ the label value $y \in \{y_{(1)}, \ldots, y_{(k)}\} \backslash \mathcal{S}_k$ that provides the maximum marginal gain $\hat{g}(\mathcal{S}_k \cup \{y\} \mid x) - \hat{g}(\mathcal{S}_k \mid x)$ until it exhausts the set $\{y_{(1)}, \ldots, y_{(k)}\}$. Moreover, at each iteration and round, it keeps track of the set with the highest objective value. At each of the $L$ runs of the greedy algorithm, at most $L$ elements are added to the set $\mathcal{S}$, and adding each element needs at most $L$ times computing the marginal gain $\hat{g}(\mathcal{S} \cup \{y\} \mid x) - \hat{g}(\mathcal{S} \mid x)$, which takes $O(L)$ to compute. Hence, our algorithm has an overall complexity of $O(L^4)$. See Appendix B for a detailed running time analysis. Algorithm 1 provides a pseudocode implementation of the procedure.

Importantly, the prediction sets provided by the greedy algorithm are guaranteed to achieve higher objective value $\hat{g}$ than those provided by any conformal predictor using a non-conformity score $s(x, y)$ that is nonincreasing with respect to $f_y(x)$, as formalized by the following proposition:[10]

**Proposition 1** *For any $x \in \mathcal{X}$, let $\mathcal{S}$ be the prediction set provided by Algorithm 1 and $\mathcal{S}_{cp}$ be the prediction set provided by any conformal prediction with a non-conformity score $s(x, y)$ that is nonincreasing with respect to $f_y(x)$, then, it holds that $\hat{g}(\mathcal{S} \mid x) \geq \hat{g}(\mathcal{S}_{cp} \mid x)$.*

# 5 Experiments with Synthetic Data

In this section, we compare the average accuracy achieved by different simulated human experts using prediction sets constructed with our greedy algorithm (Algorithm 1), brute force search, and conformal prediction on several synthetic multiclass classification tasks where the experts and the classifier used by the greedy algorithm, brute force search, and conformal prediction achieve different accuracies on their own.

**Experimental setup**. We create several synthetic multiclass classification tasks, each with $n = 20$ features per sample and varying difficulty. Out of 20 features per sample, only $d = 4$ of these features are *informative*[11] while the rest are drawn at random. Refer to Appendix C for more details about the classification tasks. For each classification task, we generate 19,000 samples, which we split into a training set (16,000 samples), a calibration set (1000 samples), a validation set (1000 samples) and a test set (1000 samples).

We use the first half of the samples in the training set to train a multinomial logistic regression model $f(x)$. This model is used by the greedy algorithm, brute force search and conformal prediction. It achieves a different average test accuracy $P(Y' = Y)$, depending on the difficulty of the classification task. We use the second half of the samples in the training set to train another multinomial logistic regression model $\hat{f}(x)$. However, during the training of this model, we modify the value $a$ of one of the (informative) features of each training sample to $(1 - \gamma)a + \gamma\epsilon$, where $\epsilon \sim \mathcal{N}(0, 1)$ and $\gamma \in [0, 1]$ controls the average accuracy of the resulting model. Then, we use the (estimated) confusion matrix $C(\gamma)$ of the predictions made by $\hat{f}(x)$ to model (the predictions made by) the simulated expert by means of a mixture of MNLs, *i.e.*, $P_\mathcal{S}(\hat{Y} = y \mid X = x, Y = y) = \frac{C_{yy}(\gamma)}{\sum_{y' \in \mathcal{S}} C_{y'y}(\gamma)}$.

Further, we use the calibration set to calibrate the (softmax) outputs of the logistic regression model $f$ using top-$k$-label calibration [52] with $k = 5$. We also use it to estimate the confusion matrices $C(\gamma)$ that parameterize the mixture of MNLs used to model (the predictions made by) the simulated human expert, and calculate the quantile $\hat{q}_\alpha$ used by conformal prediction. Finally, we use the test set to evaluate the average accuracy achieved by the simulated expert using prediction sets constructed with our greedy algorithm, brute force search and conformal prediction. Here, note that

---

[10] Proposition 1 can be generalized to conformal predictors with any non-conformity score as long as the ranking that our greedy algorithm uses is the same as the ranking induced by the non-conformity scores.

[11] A feature is informative if its value correlates with the label value.

Table 1: Empirical average test accuracy achieved by four different (simulated) human experts, each with a different noise value $\gamma$, on their own (NONE) and using prediction sets constructed with conformal prediction (NAIVE, APS, RAPS and SAPS) and with the greedy algorithm (GREEDY) on four synthetic classification tasks. In each classification task, the classifier $f$ used by conformal prediction and the greedy algorithm achieves a different average accuracy $P(Y' = Y)$. The number of labels is $L = 10$, the size of the calibration set is $m = 1000$, and we do not include brute force search because it achieves the same performance as the greedy algorithm. Each cell shows the average and standard deviation over 10 runs. We denote the best results for each classification task in bold.

| $\gamma$ | METHOD | $P(Y' = Y) = 0.3$ | $P(Y' = Y) = 0.5$ | $P(Y' = Y) = 0.7$ | $P(Y' = Y) = 0.9$ |
|---|---|---|---|---|---|
| 0.3 | NAIVE | $0.340_{\pm0.014}$ | $0.588_{\pm0.015}$ | $0.799_{\pm0.013}$ | $0.944_{\pm0.006}$ |
| | APS | $0.341_{\pm0.013}$ | $0.587_{\pm0.013}$ | $0.804_{\pm0.015}$ | $0.941_{\pm0.006}$ |
| | RAPS | $0.341_{\pm0.013}$ | $0.587_{\pm0.013}$ | $0.804_{\pm0.014}$ | $0.941_{\pm0.006}$ |
| | SAPS | $0.340_{\pm0.015}$ | $0.585_{\pm0.012}$ | $0.804_{\pm0.015}$ | $0.940_{\pm0.008}$ |
| | GREEDY | $\mathbf{0.364_{\pm0.015}}$ | $\mathbf{0.605_{\pm0.014}}$ | $\mathbf{0.824_{\pm0.012}}$ | $\mathbf{0.953_{\pm0.005}}$ |
| | NONE | $0.281_{\pm0.018}$ | $0.485_{\pm0.019}$ | $0.693_{\pm0.018}$ | $0.883_{\pm0.008}$ |
| 0.5 | NAIVE | $0.328_{\pm0.014}$ | $0.564_{\pm0.012}$ | $0.774_{\pm0.014}$ | $0.932_{\pm0.007}$ |
| | APS | $0.329_{\pm0.012}$ | $0.565_{\pm0.010}$ | $0.787_{\pm0.013}$ | $0.932_{\pm0.008}$ |
| | RAPS | $0.330_{\pm0.012}$ | $0.566_{\pm0.010}$ | $0.787_{\pm0.013}$ | $0.932_{\pm0.008}$ |
| | SAPS | $0.329_{\pm0.013}$ | $0.563_{\pm0.008}$ | $0.787_{\pm0.014}$ | $0.932_{\pm0.009}$ |
| | GREEDY | $\mathbf{0.353_{\pm0.015}}$ | $\mathbf{0.587_{\pm0.010}}$ | $\mathbf{0.805_{\pm0.012}}$ | $\mathbf{0.945_{\pm0.004}}$ |
| | NONE | $0.261_{\pm0.016}$ | $0.446_{\pm0.013}$ | $0.644_{\pm0.019}$ | $0.843_{\pm0.011}$ |
| 0.7 | NAIVE | $0.319_{\pm0.012}$ | $0.534_{\pm0.013}$ | $0.737_{\pm0.013}$ | $0.908_{\pm0.006}$ |
| | APS | $0.320_{\pm0.008}$ | $0.542_{\pm0.012}$ | $0.759_{\pm0.015}$ | $0.913_{\pm0.008}$ |
| | RAPS | $0.320_{\pm0.008}$ | $0.542_{\pm0.012}$ | $0.760_{\pm0.015}$ | $0.914_{\pm0.008}$ |
| | SAPS | $0.319_{\pm0.009}$ | $0.534_{\pm0.012}$ | $0.760_{\pm0.014}$ | $0.915_{\pm0.009}$ |
| | GREEDY | $\mathbf{0.345_{\pm0.011}}$ | $\mathbf{0.573_{\pm0.010}}$ | $\mathbf{0.784_{\pm0.013}}$ | $\mathbf{0.938_{\pm0.006}}$ |
| | NONE | $0.238_{\pm0.011}$ | $0.380_{\pm0.015}$ | $0.540_{\pm0.018}$ | $0.716_{\pm0.013}$ |
| 1.0 | NAIVE | $0.314_{\pm0.015}$ | $0.517_{\pm0.011}$ | $0.714_{\pm0.015}$ | $0.894_{\pm0.012}$ |
| | APS | $0.316_{\pm0.013}$ | $0.525_{\pm0.009}$ | $0.733_{\pm0.014}$ | $0.895_{\pm0.010}$ |
| | RAPS | $0.316_{\pm0.013}$ | $0.525_{\pm0.009}$ | $0.734_{\pm0.015}$ | $0.896_{\pm0.010}$ |
| | SAPS | $0.315_{\pm0.014}$ | $0.517_{\pm0.011}$ | $0.734_{\pm0.015}$ | $0.896_{\pm0.009}$ |
| | GREEDY | $\mathbf{0.348_{\pm0.013}}$ | $\mathbf{0.567_{\pm0.011}}$ | $\mathbf{0.782_{\pm0.015}}$ | $\mathbf{0.936_{\pm0.007}}$ |
| | NONE | $0.214_{\pm0.017}$ | $0.303_{\pm0.014}$ | $0.382_{\pm0.021}$ | $0.452_{\pm0.021}$ |

our greedy algorithm and brute force search have access to the true mixtures of MNLs used to model the simulated human expert. In our experiments, we implement conformal prediction using several non-conformity scores:

$$s(x,y) = 1 - f_y(x) \text{ (NAIVE, [20])}, \quad s(x,y) = \sum_{y':f_{y'}(x) \leq f_y(x)} f_{y'}(x) \text{ (APS, [53])},$$

$$s(x,y) = f_y(x) + \sum_{y':f_{y'}(x) \leq f_y(x)} f_{y'}(x) + \lambda_{raps} \left( o(x,y) - k_{reg} \right)^+ \quad \text{(RAPS, [49])},$$

$$s(x,y) = \begin{cases} \max_{y'} f_{y'}(x) & o(x,y) = 1 \\ \max_{y'} f_{y'}(x) + \lambda_{saps} \left( o(x,y) - 2 \right) & o(x,y) > 1 \end{cases} \quad \text{(SAPS, [54])},$$

where $o(x,y) = |\{y' : f_{y'}(x) \leq f_y(x)\}|$ denotes the ranking of label $y$ according to $f_y(x)$ and we decided to omit the randomization for APS, RAPS and SAPS as it is only required to achieve exact $1 - \alpha$ coverage and it did not have an influence on the empirical average accuracy achieved by the simulated human experts in our experiments. For RAPS and SAPS, we run the procedure outlined in Appendix E in Angelopoulous et al. [49] to optimize the additional hyperparameters, $k_{regs}$ and $\lambda_{raps}$, for RAPS, and $\lambda_{saps}$ for SAPS, using the validation set. Further, for each non-conformity score and classification task, we report the results for the $\alpha$ value under which the expert achieves the highest average test accuracy and, to avoid empty sets, we always include the label value with the lowest non-conformity score in the prediction sets. In this context, note that, in practice, one would need to select $\alpha$ using a held-out dataset, however, our evaluation aims to show how our greedy algorithm

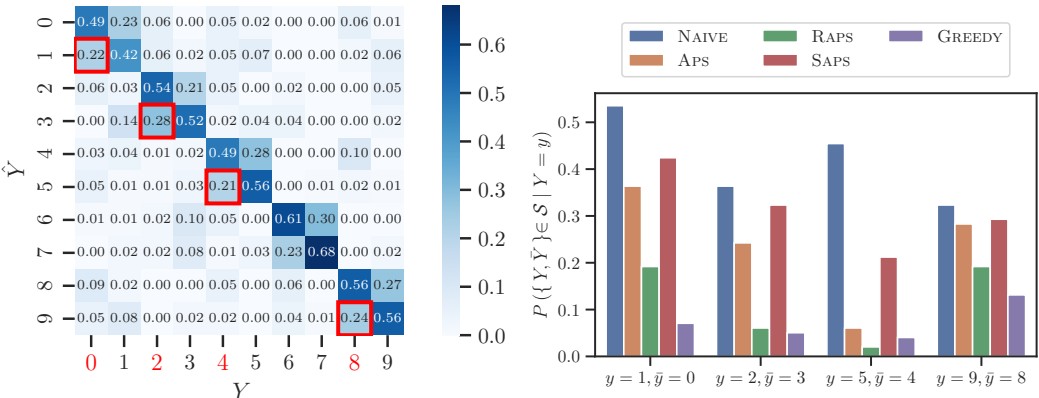

Figure 2: (Left) Confusion matrix $C$ for the predictions made by a (simulated) human expert on their own. The label $\bar{y} = \mathrm{argmax}_{y' \neq y} C_{y'y}$ that is most frequently mistaken with the ground truth-label $y$ is highlighted in red for $y \in \{0, 2, 6, 8\}$. (Right) Empirical conditional probability that a prediction set includes $\{y, \bar{y}\}$ given $Y = y$ with conformal prediction (NAIVE, APS, RAPS and SAPS) and our greedy algorithm (GREEDY). In both panels, $\gamma = 0.7$ and $\mathbb{P}(Y' = Y) = 0.7$.

improves over conformal prediction for *any* value of $\alpha$. Finally, we repeat each experiment ten times and, each time, we sample different training, calibration, validation and test sets.

**Results.** We first estimate the average test accuracy achieved by four different (simulated) human experts, each with a different $\gamma$ value, on four classification tasks where the classifier $f$ achieves a different average accuracy $P(Y' = Y)$. We report their average test accuracy on their own (NONE) and when using prediction sets constructed with conformal prediction (NAIVE, APS, RAPS and SAPS), our greedy algorithm (GREEDY) and brute force search (BRUTE FORCE SEARCH). Table 1 summarizes the results, where we have not included brute force search because it achieves the same performance as our greedy algorithm. The results show that, using the greedy algorithm to construct prediction sets, the experts consistently achieve the highest average accuracy across classification tasks. Moreover, the results also show that, under the prediction sets constructed using the greedy algorithm, the average accuracy achieved by the expert degrades gracefully as $\gamma$ increases whereas, under the prediction sets constructed using conformal prediction, the average accuracy degrades significantly. Refer to Appendix D for additional results for $L \in \{25, 50\}$ showing that the relative gain in average accuracy offered by the greedy algorithm increases with the number of labels and noise $\gamma$. Refer to Appendix E for additional results showing that the empirical average coverage achieved by the prediction sets constructed using conformal prediction and our greedy algorithm may be a bad proxy for estimating the average accuracy achieved by human experts using prediction sets.

To better understand why the prediction sets constructed by the greedy algorithm help human experts achieve higher average accuracy than those constructed by conformal prediction, we now look closer into the structure of the prediction sets. Given a ground truth-label $Y = y$, let $\bar{y} = \mathrm{argmax}_{y' \neq y} C_{y'y}$ be the label that is most frequently mistaken with $y$. Then, we estimate the empirical conditional probability that a prediction set includes $\{y, \bar{y}\}$ given $Y = y$ with the greedy algorithm and conformal prediction. Figure 2 summarizes the results for $\gamma = 0.7$ and $P(Y' = Y) = 0.7$ and $y \in \{0, 2, 6, 8\}$. Appendix F includes additional results for other configurations. The results show that, with the greedy algorithm, the empirical probability that a prediction set includes $\{y, \bar{y}\}$ given $Y = y$ is much lower (*i.e.*, 2-3x lower) than with conformal prediction despite it creates overall larger prediction sets.

# 6 Experiments with Real Data

In this section, we compare the average accuracy achieved by a simulated human expert using prediction sets constructed with our greedy algorithm, brute force search and conformal prediction on a real multiclass classification task over noisy natural images. The simulated human expert follows

Table 2: Average test accuracy achieved by a (simulated) human expert on their own (NONE), and using the prediction sets constructed with conformal prediction (NAIVE, APS, RAPS and SAPS), and our greedy algorithm (GREEDY) on the ImageNet16H dataset. We do not include brute force search because it achieves the same performance as the greedy algorithm. Each cell shows the average and standard deviation over 10 runs. We denote the best results in bold.

| METHOD | $\omega = 80$ | $\omega = 95$ | $\omega = 110$ | $\omega = 125$ |
|---|---|---|---|---|
| NAIVE | **0.957** ±0.006 | 0.946 ±0.008 | 0.919 ±0.006 | 0.860 ±0.008 |
| APS | 0.944 ±0.004 | 0.932 ±0.005 | 0.902 ±0.008 | 0.852 ±0.010 |
| RAPS | 0.950 ±0.009 | 0.943 ±0.010 | 0.914 ±0.010 | 0.849 ±0.011 |
| SAPS | 0.953 ±0.010 | 0.942 ±0.009 | 0.918 ±0.009 | 0.855 ±0.007 |
| GREEDY | **0.957** ±0.007 | **0.951** ±0.009 | **0.925** ±0.008 | **0.874** ±0.009 |
| NONE | 0.900 ±0.002 | 0.859 ±0.003 | 0.771 ±0.005 | 0.603 ±0.007 |

the mixture of MNLs introduced in Eq. 4, which is parameterized by the (estimated) confusion matrix of the predictions made by real human experts on their own.[12]

**Experimental setup.** We experiment with the ImageNet16H dataset [7], which was created using 1,200 natural images from the ImageNet Large Scale Visual Recognition Challenge (ILSRVR) 2012 dataset [55]. More specifically, in the ImageNet16H dataset, each of the above images was used to create four noisy images with different levels of phase noise distortion $\omega \in \{80, 95, 110, 125\}$ and the same ground-truth label $y$ from a label set $\mathcal{Y}$ of size $n = 16$. In addition, for each noisy image, the dataset contains (approximately) six label predictions made by human experts on their own. In our experiments, we run and evaluate each method separately by grouping the above noisy images (and expert predictions) according to their level of noise. For each group of images and method, we use the deep neural network classifier VGG-19 [56] after 10 epochs of fine-tuning as provided by Steyvers et al. [7]. Further, we randomly split the images (and expert predictions) in each group into two disjoint subsets, a calibration set (800 images), and a test set (400 images). The accuracy of the (pretrained) VGG-19 classifier on the test set is $0.900 \pm 0.014$ ($\omega = 80$), $0.895 \pm 0.009$ ($\omega = 95$), $0.857 \pm 0.016$ ($\omega = 110$) and $0.792 \pm 0.016$ ($\omega = 125$). We use the calibration set to (i) calibrate the (softmax) outputs of the VGG-19 scores using top-k-label calibration with $k = 5$, (ii) estimate the confusion matrix $\mathbf{C}$ that parameterizes the mixture of MNLs used to model the simulated human expert, and (iii) calculate the quantile $\hat{q}_\alpha$ used by conformal prediction[13].We use the test set to evaluate the average accuracy the simulated expert achieves using prediction sets constructed with our greedy algorithm, brute force search and conformal prediction. Here, note that, similarly to in the experiments in synthetic data, our greedy algorithm and brute force search have access to the true mixture of MNLs used to model the simulated human expert. We implement conformal prediction using the same non-conformity scores used in the experiments with synthetic data and, for each non-conformity score and group of images, we report the results for the $\alpha$ value under which the expert achieves the highest average test accuracy. In Appendix H, we report results for all $\alpha$ values. To obtain error bars, we repeat each experiment 10 times, sampling different calibration and test sets.

**Results.** Table 2 and Figure 3 show the average test accuracy and complementary cumulative distribution (cCDF) of the test per-image accuracy achieved by a simulated human expert using the prediction sets constructed with conformal prediction (NAIVE, APS, RAPS and SAPS), our greedy algorithm (GREEDY) and brute force search (BRUTE FORCE SEARCH) for different values of noise $\omega$. The results show that, similarly as in our experiments with synthetic data, the greedy algorithm achieves the same performance as brute force search. Moreover, they also show that, using greedy algorithm to construct prediction sets, the expert achieves the highest average accuracy in all groups of images except the group with $\omega = 80$, where both the greedy algorithm and one of the conformal predictors offer comparable performance. Similarly as in the synthetic experiments, refer to Appendix E for additional results regarding the empirical average coverage achieved by the prediction sets constructed using conformal prediction and our greedy algorithm.

---

[12]In Appendix G, we evaluate the goodness of fit of the mixture of MNLs to predictions made by real human experts using a support system based on prediction sets [19].

[13]For RAPS and SAPS, we further split the calibration set to obtain a (reduced) calibration (400 images) and validation (400 images) sets to calculate the quantile $\hat{q}_\alpha$ and optimize the hyperparameters of RAPS and SAPS, respectively.

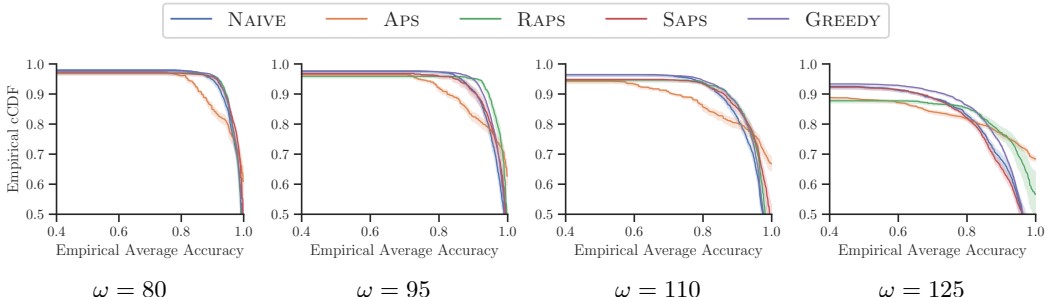

Figure 3: Complementary cumulative distribution (cCDF) of the per-image test accuracy achieved by a simulated human expert using the prediction sets constructed with conformal prediction (NAIVE, APS, RAPS and SAPS) and our greedy algorithm (GREEDY) on the ImageNet16H dataset.

## 7    Discussion and Limitations

In this section, we discuss several assumptions and limitations of our work, which open up interesting avenues for future work.

**Hardness analysis.** In our hardness analysis, our reduction utilizes an instance of our problem in which, for every prediction set, the predictions made by experts follow a mixture of MNLs. As an immediate consequence, this implies that, in general, the problem of finding the prediction set $\mathcal{S}$ that maximizes $\hat{g}(\mathcal{S} \,|\, x)$ is NP-hard to approximate. However, in our experiments, the greedy algorithm is almost always able to find such a set $\mathcal{S}$. As a result, we hypothesize that there may be certain conditions on the parameters of the mixture of MNLs under which the problem can be efficiently approximated to a factor less than the size of the label set.

**Methodology.** Our greedy algorithm assumes that, for every prediction set, the predictions made by the human expert follow a parameterized expert model—the above mentioned mixture of MNLs. It would be worthy to develop model-free algorithms since, in the context of prediction sets constructed using conformal prediction, they have been shown to be superior to their model-based counterparts [19]. To this end, a good starting point may be the literature on (contextual) combinatorial multi-armed bandits [57, 58], where one can map each arm to a label value and each subset of arms, namely a super arm, to a prediction set.

**Evaluation.** The results of our experiments suggest that the prediction sets constructed using our greedy algorithm may help human experts make more accurate predictions than the prediction sets constructed using conformal prediction. However, one may argue that the difference in performance is partly due to the fact that the non-conformity scores used in conformal prediction do not incorporate information about the distribution of experts' predictions. Motivated by this observation, it would be important to investigate how to incorporate such information in the definition of non-conformity scores. Moreover, in our experiments, the true distribution of experts' predictions matches the mixture of MNLs used by the greedy algorithm and brute force search. However, in practice, there may be a mismatch between the true distribution of experts' predictions and the mixture of MNLs, and this may decrease performance. Finally, in our experiments with real data, the ground truth labels are estimated by aggregating (multiple) predictions by human annotators using majority voting, however, this may introduce additional sources of errors that may influence our results [59].

**Broader impact.** We have focused on maximizing the average accuracy of the predictions made by an expert using a decision support system based on prediction sets. However, in high-stakes application domains, it would be important to extend our methodology to account for fairness considerations.

## 8    Conclusions

We have looked at the problem of finding the optimal prediction sets under which human experts achieve the highest accuracy in a given multiclass classification task. We have shown that this problem is NP-hard to solve and to approximate to any factor less than the size of the label set. However, we have empirically shown that, for a large parameterized class of expert models, a simple greedy algorithm consistently outperforms conformal prediction.

## Acknowledgments and Disclosure of Funding

Gomez-Rodriguez acknowledges support from the European Research Council (ERC) under the European Union's Horizon 2020 research and innovation programme (grant agreement No. 945719). De Toni acknowledges support from the TANGO project (grant #101120763-TANGO). Views and opinions expressed are however those of the author(s) only and do not necessarily reflect those of the EU or HaDEA. Neither the EU nor the granting authority can be held responsible for them.

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

# A Proofs

## A.1 Proof of Proposition 1

We prove by contradiction. Assume there exist $x \in \mathcal{X}$ such that $\hat{g}(\mathcal{S} \mid x) < \hat{g}(\mathcal{S}_{\mathrm{cp}} \mid x)$. Let $k := |\mathcal{S}_{\mathrm{cp}}|$ and $\mathcal{S}_k^*$ be the set providing the highest objective among all the sets seen at the $k$-th round of the greedy algorithm. It should hold that:

$$\hat{g}(\mathcal{S}_{\mathrm{cp}} \mid x) \underset{(i)}{=} \hat{g}(\{y_{(1)}, \ldots, y_{(k)}\} \mid x) \underset{(ii)}{\leq} \hat{g}(\mathcal{S}_k^* \mid x) \underset{(iii)}{\leq} \hat{g}(\mathcal{S} \mid x),$$

which is a contradiction, hence, it should hold that $\hat{g}(\mathcal{S} \mid x) \geq \hat{g}(\mathcal{S}_{\mathrm{cp}} \mid x)$. Note that (i) is due to the fact that the ranking imposed by the non-conformity scores are the same as the ranking considered by our greedy algorithm so whenever the conformal prediction outputs a set of size $k$, the $k$ elements correspond to same $k$ elements that are considered in the $k$-th round in the greedy algorithm, *i.e.*, $\{y_{(1)}, \ldots, y_{(k)}\}$; (ii) is due to the fact that $\mathcal{S}_k^*$ is the best set at the $k$-th round of the greedy algorithm; and (iii) is because $\mathcal{S} = \mathrm{argmax}_{\mathcal{S}_k^* \in \{\mathcal{S}_1^*, \ldots, \mathcal{S}_L^*\}} \hat{g}(\mathcal{S}_k^* \mid x)$. ∎

## A.2 Proof of Theorem 1

To establish NP-hardness we reduce an instance $\langle \mathcal{G} = (\mathcal{V}, \mathcal{E}), k \rangle$ of the $k$-clique problem,[14] which is known to be NP-complete [22], to an instance $\langle x, \mathcal{Y}, B, \mathbf{C} \rangle$ of deciding whether there exists a prediction set $\mathcal{S} \subseteq \mathcal{Y}$ such that $g(\mathcal{S} \mid x) \geq B$ given a constant $B > 0$, as follows: $x \in \mathbb{R}^d$, $\mathcal{Y} = \mathcal{V}$, $B = \frac{k}{|\mathcal{V}|}$ and, for all $y \in \mathcal{Y}$, we have that $P(Y = y \mid X = x) = \frac{1}{|\mathcal{V}|}$ and

$$P_{\mathcal{S}}(\hat{Y} = \hat{y} \mid X = x, Y = y) = \frac{C_{\hat{y}y}}{\sum_{y' \in \mathcal{S}} C_{y'y}} \quad \text{where } C_{y'y} = \begin{cases} 0 & \text{if } (y', y) \in \mathcal{E} \\ 1/\widehat{N}_{\mathcal{V}}(y) & \text{otherwise,} \end{cases}$$

and $\widehat{N}_{\mathcal{V}}(y)$ denotes the number of vertices that are not adjacent to $y$ in $\mathcal{G}$. For any $\mathcal{S} \subseteq \mathcal{Y}$,

$$g(\mathcal{S} \mid x) = \sum_{y \in \mathcal{S}} f_y(x) \frac{C_{yy}}{\sum_{y' \in \mathcal{S}} C_{y'y}} = \frac{1}{|\mathcal{Y}|} \sum_{y \in \mathcal{S}} \frac{1}{\widehat{N}_{\mathcal{S}}(y)},$$

and $\widehat{N}_{\mathcal{S}}(y)$ denotes the number of vertices that are not adjacent to $y$ in the subgraph induced[15] by $\mathcal{S}$. The transformation described above is dominated by the size of $C$, which is $O(|\mathcal{V}|^2)$, so the reduction is polynomial time.

We note that the above decision problem can be reduced, in polynomial time, to the problem of finding the optimal prediction set. Precisely, given the optimal prediction set $\mathcal{S}^*$, for every $B \leq g(\mathcal{S}^* \mid x)$ we return YES, otherwise NO. Thus, establishing the NP-hardness for the decision problem suffices.

($\Longleftarrow$) Here, we show that, if the (decision-variant of the) optimal prediction sets problem is a YES instance then the $k$-clique problem is also a YES instance. Since the optimal prediction sets problem is a YES instance, we have a subset $\mathcal{S} \subseteq \mathcal{Y}$ such that $g(\mathcal{S} \mid x) \geq \frac{k}{|\mathcal{Y}|}$. We anticipate two possibilities, either $\max_{y \in \mathcal{S}} \widehat{N}_{\mathcal{S}}(y) = 1$ or $\max_{y \in \mathcal{S}} \widehat{N}_{\mathcal{S}}(y) > 1$, in both cases, we will show that there exists a clique of size at least $k$ in $\mathcal{G}$.

If $\max_{y \in \mathcal{S}} \widehat{N}_{\mathcal{S}}(y) = 1$ then every pair of vertices $y, y' \in \mathcal{S}$ are adjacent. Since $g(\mathcal{S} \mid x) \geq \frac{k}{|\mathcal{Y}|}$, the size of $\mathcal{S}$ must be at least $k$ otherwise $g(\mathcal{S} \mid x) < \frac{k}{|\mathcal{Y}|}$. So $\mathcal{S}$ induces a clique of size at least $k$ in $\mathcal{G}$.

If $\max_{y \in \mathcal{S}} \widehat{N}_{\mathcal{S}}(y) > 1$, we (iteratively) remove $y' = \arg\max_{y \in \mathcal{S}} \widehat{N}_{\mathcal{S}}(y)$ that has least number of neighbours in $\mathcal{G}_{\mathcal{S}}$, that is the subgraph induced by vertices in $\mathcal{S}$. In Lemma 1, we show that, by removing $y' = \arg\max_{y' \in \mathcal{S}} \widehat{N}_{\mathcal{S}}(y)$ from $\mathcal{S}$, it holds that $g(\mathcal{S} \setminus \{y'\} \mid x) \geq g(\mathcal{S} \mid x)$. Further, the assertion

---

[14]Given a graph $\mathcal{G} = (\mathcal{V}, \mathcal{E})$ and an integer $k \leq |\mathcal{V}|$, the $k$-clique problem seeks to decide whether there exists $\mathcal{S} \subseteq V$ with size $|\mathcal{S}| = k$ such that, for every distinct pair $u, v \in S$, there exists $(u, v) \in \mathcal{E}$. We assume that the graph $\mathcal{G}$ is simple, do not contain self loops and do not contain multiple edges between same pair of vertices.

[15]A graph $\mathcal{G}_{\mathcal{S}} = (\mathcal{S}, \mathcal{E}_{\mathcal{S}})$ is a vertex-induced subgraph in graph $\mathcal{G} = (\mathcal{V}, \mathcal{E})$, if $\mathcal{S} \subseteq \mathcal{V}$ and for every $u, v \in \mathcal{S}$, $(u, v) \in \mathcal{E}_{\mathcal{S}}$ if and only if $(u, v) \in \mathcal{E}$.

$g(\mathcal{S} \setminus \{y'\} \mid x) \geq g(\mathcal{S} \mid x)$ remains true for each iteration as we remove $y' = \arg\max_{y \in \mathcal{S}} \widehat{N}_{\mathcal{S}}(y)$ iteratively, to obtain $\mathcal{S}' \subseteq \mathcal{S}$ such that $\max_{y \in \mathcal{S}'} \widehat{N}_{\mathcal{S}'}(y) = 1$ and it holds that $g(\mathcal{S}' \mid x) \geq \frac{k}{|\mathcal{Y}|}$. The size of $\mathcal{S}'$ is at least $k$, as any smaller subset results in $g(\mathcal{S}' \mid x) < \frac{k}{|\mathcal{Y}|}$. Since $|\mathcal{S}'| \geq k$ and every $y, y' \in \mathcal{S}'$ are adjacent, $\mathcal{S}'$ induces a clique of size at least $k$.

($\Longrightarrow$) If $k$-clique is a YES instance and $\mathcal{S} \subseteq \mathcal{V}$ be a clique of size $k$, then $g(\mathcal{S} \mid x) = \frac{k}{|\mathcal{V}|} = \frac{k}{|\mathcal{Y}|}$. So the optimal predictions set problem is a YES instance. This concludes the proof. ∎

**Lemma 1** *For any subset $\mathcal{S} \subseteq \mathcal{Y}$ with $\max_{y \in \mathcal{S}} \widehat{N}(y) > 1$ and $y' = \arg\max_{y \in \mathcal{S}} \widehat{N}_{\mathcal{S}}(y)$, it holds that $g(\mathcal{S} \setminus \{y'\} \mid x) \geq g(\mathcal{S} \mid x)$.*

**Proof**

$$
g(\mathcal{S} \setminus \{y'\} \mid x) - g(\mathcal{S} \mid x) = \frac{1}{|\mathcal{Y}|} \left( \sum_{y \in \widehat{A}_{\mathcal{S} \setminus \{y'\}}(y')} \left( \frac{1}{\widehat{N}_{\mathcal{S}}(y) - 1} - \frac{1}{\widehat{N}_{\mathcal{S}}(y)} \right) - \frac{1}{\widehat{N}_{\mathcal{S}}(y')} \right)
$$

$$
= \frac{1}{|\mathcal{Y}|} \left( \sum_{y \in \widehat{A}_{\mathcal{S} \setminus \{y'\}}(y')} \left( \frac{1}{(\widehat{N}_{\mathcal{S}}(y) - 1)(\widehat{N}_{\mathcal{S}}(y))} \right) - \frac{1}{\widehat{N}_{\mathcal{S}}(y')} \right)
$$

$$
\overset{(i)}{\geq} \frac{1}{|\mathcal{Y}|} \left( \sum_{y \in \widehat{A}_{\mathcal{S} \setminus \{y'\}}(y')} \left( \frac{1}{(\widehat{N}_{\mathcal{S}}(y') - 1)(\widehat{N}_{\mathcal{S}}(y'))} \right) - \frac{1}{\widehat{N}_{\mathcal{S}}(y')} \right)
$$

$$
\overset{(ii)}{\geq} 0
$$

Note that $(i)$ and $(ii)$ are valid because $y' = \arg\max_{y \in \mathcal{S}} \widehat{N}_{\mathcal{S}}(y)$. ∎

## A.3  Proof of Theorem 2

To establish the hardness of approximation, in Lemma 2 we will show that, given a polynomial-time $\alpha$-approximation algorithm for the problem of finding the optimal prediction set, we can obtain a polynomial-time $\alpha$-approximation algorithm for the problem of finding a maximum clique [16] in a graph $\mathcal{G} = (\mathcal{V}, \mathcal{E})$. It is known that, assuming $\mathsf{P} \neq \mathsf{NP}$, for every $\epsilon > 0$, the latter problem is NP-hard to approximate to a factor $|\mathcal{V}|^{1-\epsilon}$ [51]. So we conclude that the optimal prediction sets problem is NP-hard to approximate to a factor $|\mathcal{Y}|^{1-\epsilon}$. ∎

**Lemma 2** *Suppose there exists a polynomial-time $\alpha$-approximation algorithm for the optimal prediction sets problem with $\alpha \geq 1$, then there exists a polynomial-time $\alpha$-approximation algorithm for the maximum clique problem.*

**Proof**  Let $\mathcal{S}^* \subseteq \mathcal{Y}$ denote the optimal solution for the optimal prediction sets problem, as defined in Eq 1. A subset $\mathcal{S} \subseteq \mathcal{Y}$ is an $\alpha$-approximation for the optimal prediction sets problem if $g(\mathcal{S} \mid x) \cdot \alpha \geq g(\mathcal{S}^* \mid x)$. We say an algorithm approximates an instance of the maximum clique problem within a factor $\alpha$ if it can find a clique of size at least $\lfloor \frac{k^*}{\alpha} \rfloor$, when the graph contains a max-clique of size $k^*$.

The reduction closely resembles the construction outlined in the proof of Theorem 1, with a subtly distinction being the absence of a bound $B$. For completeness, we will describe the construction again. Given an instance $\langle G = (V, E) \rangle$ of the maximum clique problem, we construct an instance $\langle x, \mathcal{Y}, \boldsymbol{C} \rangle$ of the optimal prediction sets problem as follows: $x \in \mathbb{R}^d$, $\mathcal{Y} = \mathcal{V}$ and, for all $y \in \mathcal{Y}$, we have that $P(Y = y \mid X = x) = \frac{1}{|\mathcal{V}|}$ and

$$
P_{\mathcal{S}}(\hat{Y} = \hat{y} \mid X = x, Y = y) = \frac{C_{\hat{y}y}}{\sum_{y' \in \mathcal{S}} C_{y'y}} \text{ where } C_{y'y} = \begin{cases} 0 & \text{if } (y', y) \in \mathcal{E} \\ 1/\widehat{N}_{\mathcal{V}}(y) & \text{otherwise,} \end{cases}
$$

---

[16] Given a graph $\mathcal{G} = (\mathcal{V}, \mathcal{E})$, the maximum clique problem seeks to find the largest $\mathcal{S} \subseteq \mathcal{V}$ such that, for every $u, v \in \mathcal{S}$, there exists $(u, v) \in \mathcal{E}$.

and $\widehat{N}_{\mathcal{V}}(y)$ denotes the number of vertices that are not adjacent to $y$ in $\mathcal{G}$. For any $\mathcal{S} \subseteq \mathcal{Y}$,

$$g(\mathcal{S}\,|\,x) = \sum_{y \in \mathcal{S}} f_y(x) \frac{C_{yy}}{\sum_{y' \in \mathcal{S}} C_{y'y}} = \frac{1}{|\mathcal{Y}|} \sum_{y \in \mathcal{S}} \frac{1}{\widehat{N}_{\mathcal{S}}(y)},$$

and $\widehat{N}_{\mathcal{S}}(y)$ denotes the number of vertices that are not adjacent to $y$ in the subgraph induced by $\mathcal{S}$.

Let $\mathcal{S}^* \subseteq \mathcal{V}$ be a maximum clique of size $k^*$ in $\mathcal{G}$. For all $y \in \mathcal{S}^*$, $\widehat{N}_{\mathcal{S}^*}(y) = 1$ and $g(\mathcal{S}^*\,|\,x) = \frac{k^*}{|\mathcal{Y}|}$. In Lemma 3 we show that, for any $\mathcal{S} \subseteq \mathcal{Y}$ if $\mathcal{S}$ is not a clique of size $k^*$, then $g(\mathcal{S}\,|\,x) \leq g(\mathcal{S}^*\,|\,x)$. Further, if $\mathcal{S} \subseteq \mathcal{Y}$ is an $\alpha$-approximation, then from the definition of approximation ratio, it holds that,

$$g(\mathcal{S}\,|\,x) \geq \frac{1}{\alpha} \cdot g(\mathcal{S}^*\,|\,x) \implies g(\mathcal{S}\,|\,x) \geq \lfloor \frac{k^*}{\alpha\,|\mathcal{Y}|} \rfloor.$$

In the remainder of this proof, we show that there exists a clique of size at least $\lfloor \frac{k^*}{\alpha} \rfloor$ in $\mathcal{G}$. To do so, we consider two possibilities: $\max_{y \in \mathcal{S}} \widehat{N}(y) = 1$ and $\max_{y \in \mathcal{S}} \widehat{N}(y) > 1$. In both cases, we will establish the validity of the aforementioned claim.

If $\max_{y \in \mathcal{S}} \widehat{N}_{\mathcal{S}}(y) = 1$ and $g(\mathcal{S}\,|\,x) = \frac{k^*}{\alpha|\mathcal{Y}|}$, then $|\mathcal{S}| = \frac{k^*}{\alpha}$ and every $y, y' \in \mathcal{S}$ is adjacent in the subgraph induced by $\mathcal{S}$. So $\mathcal{S}$ induces a clique of size $\frac{k^*}{\alpha}$.

If $\max_{y \in \mathcal{S}} \widehat{N}_{\mathcal{S}}(y) > 1$, we (iteratively) remove $y' = \arg\max_{y \in \mathcal{S}} \widehat{N}_{\mathcal{S}}(y)$. In Lemma 1, we show that, by removing $y' = \arg\max_{y' \in \mathcal{S}} \widehat{N}_{\mathcal{S}}(y)$ from $\mathcal{S}$, it holds that $g(\mathcal{S} \setminus \{y'\}\,|\,x) \geq g(\mathcal{S}\,|\,x)$. Further, the assertion $g(\mathcal{S} \setminus \{y'\}\,|\,x) \geq g(\mathcal{S}\,|\,x)$ remains true for each iteration as we remove $y' = \arg\max_{y \in \mathcal{S}} \widehat{N}_{\mathcal{S}}(y)$ iteratively, to obtain $\mathcal{S}' \subseteq \mathcal{S}$ such that $\max_{y \in \mathcal{S}'} \widehat{N}_{\mathcal{S}'}(y) = 1$ and it holds that $g(\mathcal{S}'\,|\,x) \geq \lfloor \frac{k^*}{\alpha|\mathcal{Y}|} \rfloor$. The size of $\mathcal{S}'$ is at least $\lfloor \frac{k^*}{\alpha} \rfloor$, as any smaller subset results in $g(\mathcal{S}'\,|\,x) < \frac{k}{\alpha|\mathcal{Y}|}$. Since $|\mathcal{S}'| \geq \lfloor \frac{k^*}{\alpha} \rfloor$ and every $y, y' \in \mathcal{S}'$ are adjacent. So, we conclude that $\mathcal{S}'$ induces a clique of size at least $\lfloor \frac{k^*}{\alpha} \rfloor$. ∎

**Lemma 3** *For any $\mathcal{S} \subseteq \mathcal{V}$, if the vertices in $\mathcal{S}$ do not induce a maximum clique in $\mathcal{G}$, then $g(\mathcal{S}\,|\,x) \leq g(\mathcal{S}^*\,|\,x)$, where $\mathcal{S}^* \subseteq \mathcal{V}$ induces a maximum clique in $\mathcal{G}$.*

**Proof** Let $|\mathcal{S}| = k$ and $|\mathcal{S}^*| = k^*$. Based on the values $k$ and $k^*$ can take, we have three possibilities: $(i)$ $k < k^*$, $(ii)$ $k = k^*$, and $(iii)$ $k > k^*$. In each case, we show that the aforementioned claim holds.

Case $(i)$: If $k < k^*$, then $g(\mathcal{S}\,|\,x) = \frac{k-1}{|\mathcal{Y}|} < g(\mathcal{S}^*\,|\,x)$.

Case $(ii)$: If $k = k^*$, then $g(\mathcal{S}\,|\,x) = \frac{k^*}{|\mathcal{Y}|} = g(\mathcal{S}^*\,|\,x)$.

Case $(iii)$: If $k > k^*$ and $\mathcal{S}$ do not induce a clique, then there exists at least one pair $y, y' \in \mathcal{S}$ that are not adjacent, and the value of $g(\mathcal{S}\,|\,x) \leq \frac{k-1}{|\mathcal{Y}|}$. Without loss of generality, assume that $g(\mathcal{S}\,|\,x) = \frac{k-1}{|\mathcal{Y}|}$. Note that, if there are more vertex pairs that are not adjacent, then the inequality is strict, *i.e.*, $g(\mathcal{S}\,|\,x) < \frac{k-1}{|\mathcal{Y}|}$. Let $\mathcal{S} \setminus \{y\}$ be a clique of size $k - 1$. In order for $g(x, \mathcal{S}) > g(x, \mathcal{S}^*)$ to hold, $\mathcal{S}$ must contain a clique of size greater than $k^* + 1$, which contradicts our premise that the maximum clique in $\mathcal{G}$ has size $k^*$. More precisely,

$$g(\mathcal{S}\,|\,x) > g(\mathcal{S}^*\,|\,x) \implies \frac{k-1}{|\mathcal{Y}|} > \frac{k^*}{|\mathcal{Y}|} \implies k > k^* + 1.$$

This concludes the proof. ∎

## B  Running time analysis of the greedy algorithm

In this section, we present the complexity analysis for the greedy algorithm in Algorithm 1. Adding each element requires computing the marginal gain $\Delta = \hat{g}(\mathcal{S} \cup \{y\} \,|\, x) - \hat{g}(\mathcal{S} \,|\, x)$ at most $k$ times. This computation in Line 8 can be efficiently performed in $O(k)$ time, where $k$ is the size of $\mathcal{S}_k$. Specifically, it can be rewritten as:

$$\hat{g}(\mathcal{S}_k \cup \{y\} \,|\, x) - \hat{g}(\mathcal{S}_k \,|\, x) = \sum_{\hat{y} \in \mathcal{S}_k \cup \{y\}} f_{\hat{y}}(x) \frac{C_{\hat{y}\hat{y}}}{\sum_{y' \in \mathcal{S}_k \cup \{y\}} C_{y'\hat{y}}} - \sum_{\hat{y} \in \mathcal{S}_k} f_{\hat{y}}(x) \frac{C_{\hat{y}\hat{y}}}{\sum_{y' \in \mathcal{S}_k} C_{y'\hat{y}}}.$$

The term $\sum_{y' \in \mathcal{S}_k \cup \{y\}} C_{y'\hat{y}}$ in the denominator can be computed as $\sum_{y' \in \mathcal{S}_k} C_{y'\hat{y}} + C_{y\hat{y}}$ by storing the value of $\sum_{y' \in \mathcal{S}_k} C_{y'\hat{y}}$ at the end of each iteration after Line 11. The loops in Line 5 and Line 7 each iterate over $k$, resulting in $O(k^3)$ iterations for each value of $k \in \{1, \ldots, L\}$. So, the overall running time of our algorithm is $O(L^4)$.

## C  Implementation details for the experiments with synthetic and real data

We report the implementation details and computational resources used to run the experiments in Section 5 and Section 6. The code infrastructure was written using Python 3.8 and the standard set of scientific opensource libraries (e.g., `numpy`, `pandas`, `scikit-learn`, etc.). The full set of requirements can be found in the released code. We run the experiment on a Linux machine equipped with an Intel® Xeon(R) Gold 6252N CPU, with 96 cores and 1024 GB of RAM. Practically, our experiments and the algorithms require little resources to be run. We parallelized the execution using `OpenMPI` and 50 physical cores and 20 GB of RAM.

**Experiments with synthetic data.** We employ the `make_classification` utility function of `scikit-learn` to generate the various prediction task. It is a convenient method to generate $L$-class classification tasks by varying several parameters such as the task difficulty, the number of labels and the number of informative features. It generates $n$ clusters of points positioned on the vertices of a $d$-dimensional hypercube by adding interdependencies and noise to the features. In Section 5, we set the number of features to 20, the number of redundant features to 0 and the number of informative features to $d = 4$ (for a $L = 10$ label classification task). We assign a balanced proportion of samples for each class. We control the task difficulty by choosing the `class_sep` parameter, which represents the length of the sides of the hypercubes, thus indicating how far apart are the various classes. A smaller `class_sep` implies a more difficult classification task. We vary the `class_sep` parameter to ensure the classifier and the humans span different ranges of accuracies. Please refer to the original documentation for more information.[17]

**Experiments with real data.** We use the data provided by Steyvers et al. [7] to evaluate our algorithm against the best conformal predictors. The dataset is composed of 1200 images from the ImageNet-16H classification task. The authors provide also a noisy version of these images by applying a different phase noise $\omega$. Noisier images imply a more difficult classification task. For each image and each phase noise, they provide also the softmax scores of several pre-trained classifiers for different levels of fine-tuning: baseline (no fine-tuning), between 0 and 1 epochs, 1 epochs and 10 epochs. We use the classifier scores (VGG-19 fine-tuned for 10 epochs) and the human classification performance alone for all the 1200 images for different levels of phase noise $\omega = \{80, 95, 110, 125\}$. We chose the VGG-19 classifier because it is the one achieving consistently higher accuracy than the expert alone in the classification task, for all phase noise levels. The data are freely available online.[18] In our experiment, we used Steyvers et al. *normalized* softmax scores which are obtained by dropping from VGG-19 all the irrelevant classes and by renormalizing.

---

[17] https://scikit-learn.org/stable/modules/generated/sklearn.datasets.make_
classification.html#sklearn.datasets.make_classification

[18] https://osf.io/2ntrf/wiki/home/

# D  Empirical average test accuracy on additional classification tasks

In this section, we keep the same experimental setting as described in Section 5 but we vary the number of labels $L \in \{25, 50\}$ of the classification task. Table 3 summarizes the results.

Table 3: Empirical average test accuracy for $L \in \{25, 50\}$.

| Noise | Method | $P(Y'=Y)=0.3$ | $P(Y'=Y)=0.5$ | $P(Y'=Y)=0.7$ | $P(Y'=Y)=0.9$ |
|---|---|---|---|---|---|
| 0.3 | NAIVE | $0.399_{\pm 0.012}$ | $0.614_{\pm 0.009}$ | $0.892_{\pm 0.011}$ | $0.947_{\pm 0.005}$ |
| | APS | $0.395_{\pm 0.011}$ | $0.601_{\pm 0.008}$ | $0.873_{\pm 0.009}$ | $0.938_{\pm 0.008}$ |
| | RAPS | $0.395_{\pm 0.011}$ | $0.601_{\pm 0.008}$ | $0.874_{\pm 0.009}$ | $0.939_{\pm 0.008}$ |
| | SAPS | $0.396_{\pm 0.010}$ | $0.601_{\pm 0.008}$ | $0.876_{\pm 0.013}$ | $0.943_{\pm 0.007}$ |
| | GREEDY | $\mathbf{0.431}_{\pm \mathbf{0.010}}$ | $\mathbf{0.649}_{\pm \mathbf{0.011}}$ | $\mathbf{0.902}_{\pm \mathbf{0.011}}$ | $\mathbf{0.956}_{\pm \mathbf{0.005}}$ |
| | NONE | $0.301_{\pm 0.012}$ | $0.490_{\pm 0.015}$ | $0.805_{\pm 0.012}$ | $0.904_{\pm 0.010}$ |
| 0.5 | NAIVE | $0.383_{\pm 0.011}$ | $0.595_{\pm 0.009}$ | $0.879_{\pm 0.011}$ | $0.941_{\pm 0.005}$ |
| | APS | $0.380_{\pm 0.010}$ | $0.585_{\pm 0.009}$ | $0.862_{\pm 0.011}$ | $0.932_{\pm 0.008}$ |
| | RAPS | $0.380_{\pm 0.010}$ | $0.585_{\pm 0.009}$ | $0.863_{\pm 0.011}$ | $0.933_{\pm 0.008}$ |
| | SAPS | $0.380_{\pm 0.009}$ | $0.585_{\pm 0.011}$ | $0.863_{\pm 0.014}$ | $0.937_{\pm 0.007}$ |
| | GREEDY | $\mathbf{0.417}_{\pm \mathbf{0.013}}$ | $\mathbf{0.634}_{\pm \mathbf{0.012}}$ | $\mathbf{0.894}_{\pm \mathbf{0.010}}$ | $\mathbf{0.952}_{\pm \mathbf{0.008}}$ |
| | NONE | $0.278_{\pm 0.011}$ | $0.455_{\pm 0.014}$ | $0.767_{\pm 0.012}$ | $0.879_{\pm 0.010}$ |
| 0.7 | NAIVE | $0.365_{\pm 0.012}$ | $0.566_{\pm 0.009}$ | $0.848_{\pm 0.010}$ | $0.923_{\pm 0.007}$ |
| | APS | $0.363_{\pm 0.009}$ | $0.560_{\pm 0.010}$ | $0.840_{\pm 0.012}$ | $0.918_{\pm 0.008}$ |
| | RAPS | $0.363_{\pm 0.009}$ | $0.560_{\pm 0.011}$ | $0.841_{\pm 0.011}$ | $0.920_{\pm 0.007}$ |
| | SAPS | $0.362_{\pm 0.009}$ | $0.558_{\pm 0.015}$ | $0.840_{\pm 0.014}$ | $0.920_{\pm 0.008}$ |
| | GREEDY | $\mathbf{0.408}_{\pm \mathbf{0.013}}$ | $\mathbf{0.621}_{\pm \mathbf{0.014}}$ | $\mathbf{0.885}_{\pm \mathbf{0.011}}$ | $\mathbf{0.944}_{\pm \mathbf{0.008}}$ |
| | NONE | $0.244_{\pm 0.012}$ | $0.391_{\pm 0.014}$ | $0.661_{\pm 0.010}$ | $0.772_{\pm 0.009}$ |
| 1.0 | NAIVE | $0.343_{\pm 0.012}$ | $0.530_{\pm 0.017}$ | $0.819_{\pm 0.012}$ | $0.906_{\pm 0.009}$ |
| | APS | $0.346_{\pm 0.011}$ | $0.529_{\pm 0.016}$ | $0.822_{\pm 0.013}$ | $0.906_{\pm 0.009}$ |
| | RAPS | $0.346_{\pm 0.011}$ | $0.529_{\pm 0.016}$ | $0.823_{\pm 0.013}$ | $0.907_{\pm 0.009}$ |
| | SAPS | $0.344_{\pm 0.010}$ | $0.528_{\pm 0.017}$ | $0.822_{\pm 0.014}$ | $0.907_{\pm 0.009}$ |
| | GREEDY | $\mathbf{0.408}_{\pm \mathbf{0.012}}$ | $\mathbf{0.618}_{\pm \mathbf{0.014}}$ | $\mathbf{0.888}_{\pm \mathbf{0.012}}$ | $\mathbf{0.949}_{\pm \mathbf{0.006}}$ |
| | NONE | $0.197_{\pm 0.006}$ | $0.290_{\pm 0.008}$ | $0.432_{\pm 0.006}$ | $0.475_{\pm 0.010}$ |

$$L = 25$$

| Noise | Method | $P(Y'=Y)=0.3$ | $P(Y'=Y)=0.5$ | $P(Y'=Y)=0.7$ | $P(Y'=Y)=0.9$ |
|---|---|---|---|---|---|
| 0.3 | NAIVE | $0.435_{\pm 0.014}$ | $0.656_{\pm 0.016}$ | $0.813_{\pm 0.015}$ | $0.939_{\pm 0.007}$ |
| | APS | $0.413_{\pm 0.018}$ | $0.626_{\pm 0.016}$ | $0.792_{\pm 0.013}$ | $0.932_{\pm 0.009}$ |
| | RAPS | $0.412_{\pm 0.017}$ | $0.625_{\pm 0.016}$ | $0.791_{\pm 0.013}$ | $0.932_{\pm 0.009}$ |
| | SAPS | $0.425_{\pm 0.020}$ | $0.635_{\pm 0.034}$ | $0.802_{\pm 0.014}$ | $0.934_{\pm 0.009}$ |
| | GREEDY | $\mathbf{0.477}_{\pm \mathbf{0.016}}$ | $\mathbf{0.685}_{\pm \mathbf{0.010}}$ | $\mathbf{0.832}_{\pm \mathbf{0.011}}$ | $\mathbf{0.956}_{\pm \mathbf{0.007}}$ |
| | NONE | $0.303_{\pm 0.012}$ | $0.508_{\pm 0.006}$ | $0.699_{\pm 0.011}$ | $0.907_{\pm 0.009}$ |
| 0.5 | NAIVE | $0.417_{\pm 0.015}$ | $0.640_{\pm 0.016}$ | $0.803_{\pm 0.013}$ | $0.937_{\pm 0.008}$ |
| | APS | $0.399_{\pm 0.018}$ | $0.613_{\pm 0.016}$ | $0.782_{\pm 0.014}$ | $0.927_{\pm 0.010}$ |
| | RAPS | $0.398_{\pm 0.018}$ | $0.612_{\pm 0.016}$ | $0.781_{\pm 0.014}$ | $0.927_{\pm 0.010}$ |
| | SAPS | $0.407_{\pm 0.021}$ | $0.620_{\pm 0.033}$ | $0.793_{\pm 0.014}$ | $0.931_{\pm 0.009}$ |
| | GREEDY | $\mathbf{0.463}_{\pm \mathbf{0.024}}$ | $\mathbf{0.674}_{\pm \mathbf{0.012}}$ | $\mathbf{0.824}_{\pm \mathbf{0.012}}$ | $\mathbf{0.953}_{\pm \mathbf{0.007}}$ |
| | NONE | $0.281_{\pm 0.011}$ | $0.477_{\pm 0.009}$ | $0.669_{\pm 0.012}$ | $0.890_{\pm 0.005}$ |
| 0.7 | NAIVE | $0.396_{\pm 0.005}$ | $0.606_{\pm 0.018}$ | $0.770_{\pm 0.011}$ | $0.925_{\pm 0.007}$ |
| | APS | $0.380_{\pm 0.010}$ | $0.587_{\pm 0.016}$ | $0.754_{\pm 0.015}$ | $0.915_{\pm 0.012}$ |
| | RAPS | $0.380_{\pm 0.010}$ | $0.587_{\pm 0.016}$ | $0.754_{\pm 0.015}$ | $0.916_{\pm 0.012}$ |
| | SAPS | $0.387_{\pm 0.008}$ | $0.590_{\pm 0.029}$ | $0.764_{\pm 0.013}$ | $0.920_{\pm 0.009}$ |
| | GREEDY | $\mathbf{0.450}_{\pm \mathbf{0.017}}$ | $\mathbf{0.657}_{\pm \mathbf{0.015}}$ | $\mathbf{0.811}_{\pm \mathbf{0.015}}$ | $\mathbf{0.944}_{\pm \mathbf{0.005}}$ |
| | NONE | $0.246_{\pm 0.009}$ | $0.409_{\pm 0.011}$ | $0.574_{\pm 0.011}$ | $0.798_{\pm 0.010}$ |
| 1.0 | NAIVE | $0.376_{\pm 0.012}$ | $0.569_{\pm 0.013}$ | $0.727_{\pm 0.014}$ | $0.907_{\pm 0.011}$ |
| | APS | $0.367_{\pm 0.014}$ | $0.560_{\pm 0.016}$ | $0.722_{\pm 0.017}$ | $0.905_{\pm 0.012}$ |
| | RAPS | $0.367_{\pm 0.014}$ | $0.560_{\pm 0.015}$ | $0.722_{\pm 0.017}$ | $0.906_{\pm 0.012}$ |
| | SAPS | $0.367_{\pm 0.017}$ | $0.556_{\pm 0.028}$ | $0.725_{\pm 0.020}$ | $0.907_{\pm 0.012}$ |
| | GREEDY | $\mathbf{0.463}_{\pm \mathbf{0.022}}$ | $\mathbf{0.665}_{\pm \mathbf{0.012}}$ | $\mathbf{0.818}_{\pm \mathbf{0.017}}$ | $\mathbf{0.946}_{\pm \mathbf{0.007}}$ |
| | NONE | $0.193_{\pm 0.011}$ | $0.296_{\pm 0.013}$ | $0.380_{\pm 0.012}$ | $0.463_{\pm 0.007}$ |

$$L = 50$$

# E  Empirical average test coverage of prediction sets across classification tasks

In this section, we report the empirical average test coverage across synthetic and real classification tasks achieved by the prediction sets constructed using conformal prediction and our greedy algorithms. Table 6 and Table 7 summarizes the results, which show that the prediction sets constructed using the best conformal predictors and our greedy algorithm achieve comparable empirical coverage. Moreover, in those few settings in which the prediction sets constructed using conformal prediction achieve higher coverage (*e.g.*, $P(Y' = Y) = 0.3$ and $\gamma = 0.3$), the average test accuracy achieved by the simulated human experts using the prediction sets constructed using conformal prediction is lower (cf. Table 1), which underlines how coverage alone may be a bad proxy for estimating the average accuracy achieved by human experts using prediction sets.

Table 6: Empirical average test coverage achieved by the prediction sets constructed using conformal prediction (NAIVE, APS, RAPS, SAPS) and using the greedy algorithm (GREEDY) on four synthetic classification tasks with four different (simulated) human experts, each with a different noise value $\gamma$. For each classification task, the classifier $f$ used by conformal prediction and the greedy algorithm achieves a different average accuracy $P(Y' = Y)$. For each (simulated) human expert, the best value of $\alpha$ for each conformal predictor is different and thus the reported coverage is different. The number of labels is $L = 10$ and, the size of the calibration set is $m = 1000$. Each cell shows the average and standard deviation over 10 runs. We denote the best results for each task in bold.

| $\gamma$ | **METHOD** | $\mathbb{P}[Y'=Y]=0.3$ | $\mathbb{P}[Y'=Y]=0.5$ | $\mathbb{P}[Y'=Y]=0.7$ | $\mathbb{P}[Y'=Y]=0.9$ |
|---|---|---|---|---|---|
| | NAIVE | **0.637** ±0.066 | 0.802 ±0.058 | 0.908 ±0.020 | 0.973 ±0.007 |
| | APS | 0.603 ±0.083 | **0.804** ±0.045 | 0.900 ±0.026 | 0.967 ±0.006 |
| 0.3 | RAPS | 0.592 ±0.087 | 0.792 ±0.032 | 0.911 ±0.015 | 0.965 ±0.012 |
| | SAPS | 0.580 ±0.067 | 0.794 ±0.041 | 0.890 ±0.013 | 0.965 ±0.010 |
| | GREEDY | 0.502 ±0.024 | 0.764 ±0.019 | **0.920** ±0.012 | **0.976** ±0.004 |
| | NAIVE | **0.583** ±0.104 | **0.770** ±0.044 | 0.897 ±0.021 | 0.968 ±0.009 |
| | APS | 0.557 ±0.100 | 0.741 ±0.036 | 0.879 ±0.016 | 0.961 ±0.007 |
| 0.5 | RAPS | 0.534 ±0.096 | 0.761 ±0.040 | 0.844 ±0.043 | 0.950 ±0.012 |
| | SAPS | 0.557 ±0.081 | 0.768 ±0.042 | 0.876 ±0.014 | 0.961 ±0.009 |
| | GREEDY | 0.489 ±0.027 | 0.732 ±0.015 | **0.902** ±0.013 | **0.970** ±0.003 |
| | NAIVE | **0.535** ±0.104 | 0.676 ±0.047 | 0.823 ±0.044 | 0.938 ±0.013 |
| | APS | 0.519 ±0.115 | 0.661 ±0.036 | 0.853 ±0.015 | 0.941 ±0.013 |
| 0.7 | RAPS | 0.506 ±0.071 | 0.644 ±0.056 | 0.813 ±0.019 | 0.938 ±0.010 |
| | SAPS | 0.492 ±0.079 | **0.721** ±0.051 | 0.858 ±0.031 | 0.942 ±0.011 |
| | GREEDY | 0.473 ±0.017 | 0.696 ±0.014 | **0.861** ±0.014 | **0.958** ±0.005 |
| | NAIVE | **0.499** ±0.075 | 0.608 ±0.034 | 0.750 ±0.025 | 0.905 ±0.014 |
| | APS | 0.453 ±0.062 | 0.631 ±0.038 | 0.806 ±0.026 | 0.912 ±0.013 |
| 1.0 | RAPS | 0.466 ±0.071 | 0.611 ±0.024 | 0.787 ±0.026 | 0.920 ±0.013 |
| | SAPS | 0.484 ±0.070 | 0.609 ±0.063 | 0.764 ±0.022 | 0.907 ±0.006 |
| | GREEDY | 0.457 ±0.024 | **0.664** ±0.015 | **0.839** ±0.014 | **0.951** ±0.006 |

Table 7: Empirical coverage achieved by the prediction sets constructed using conformal prediction (NAIVE, APS, RAPS, SAPS) and using the greedy algorithm (GREEDY) on the ImageNet-16H dataset. Each cell shows the average and standard deviation over 10 runs. We denote the best results for each noise level in bold.

| METHOD | $\omega = 80$ | $\omega = 95$ | $\omega = 110$ | $\omega = 125$ |
|---|---|---|---|---|
| NAIVE | **0.977** ±0.005 | **0.972** ±0.009 | **0.962** ±0.010 | 0.923 ±0.018 |
| APS | 0.970 ±0.009 | 0.962 ±0.007 | 0.943 ±0.015 | 0.887 ±0.009 |
| RAPS | 0.967 ±0.009 | 0.956 ±0.007 | 0.944 ±0.013 | 0.876 ±0.015 |
| SAPS | 0.967 ±0.009 | 0.966 ±0.009 | 0.946 ±0.008 | 0.921 ±0.011 |
| GREEDY | 0.975 ±0.008 | **0.972** ±0.010 | **0.962** ±0.008 | **0.931** ±0.007 |

# F   Empirical conditional probability that a prediction set includes $\{y, \bar{y}\}$ given a ground-truth label $Y = y$ for additional classification tasks

Given a ground truth-label $Y = y$, let $\bar{y} = \operatorname{argmax}_{y' \neq y} C_{y'y}$ be the label that is most frequently mistaken with $y$. Then, we estimate the empirical conditional probability that a prediction set includes $\{y, \bar{y}\}$ given $Y = y$ with the greedy algorithm and conformal prediction. Figure 4 summarizes the results for different $\gamma$ and $P(Y' = Y)$ values of several classification tasks where the classifier $f$ and the simulated human expert achieve different average accuracy on their own. The results show that, as the average accuracy achieved by the expert on their own worsens ($\gamma$ increases), the empirical probability that a prediction set constructed by the greedy algorithm includes $\{y, \bar{y}\}$ decreases.

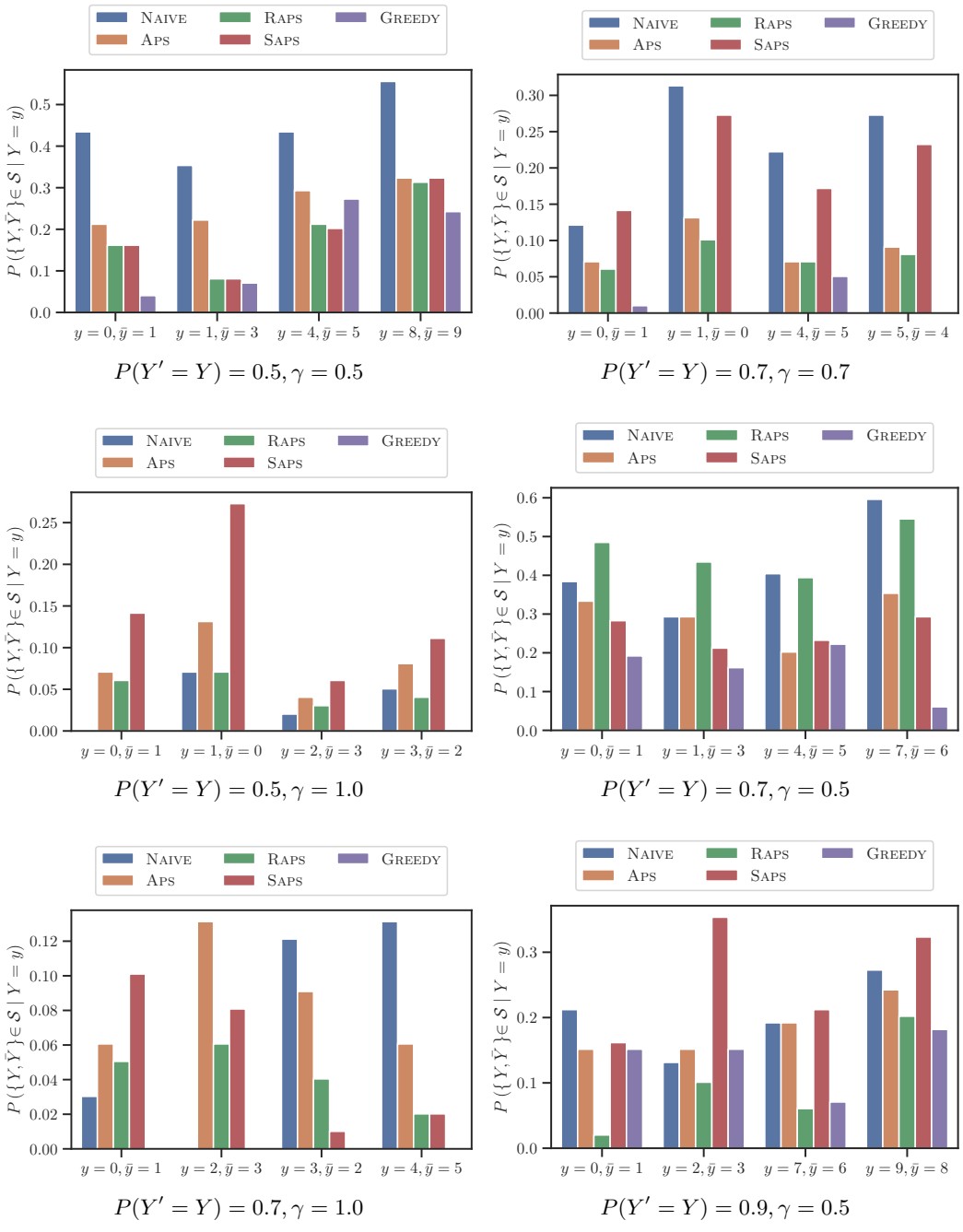

Figure 4: Empirical conditional probability that a prediction set includes $\{y, \bar{y}\}$ given $Y = y$ with conformal prediction (NAIVE, APS, RAPS and SAPS) and our greedy algorithm (GREEDY).

# G    Evaluation of the Mixture of Multinomial Logit Models (MNLs)

For the group of images with $\omega = 110$ from the ImageNet16H dataset, Straitouri et al. [19] have gathered predictions made by real human experts using prediction sets constructed by all possible conformal predictors with the first non-conformity score we have considered in our experiments (NAIVE), given a choice of calibration set. Here, we use these predictions to evaluate the goodness of fit of the mixture of MNLs used in our experiments.

Figure 5 shows that the average accuracy achieved by a simulated expert following the mixture of MNLs and by real human experts using the prediction sets constructed with all possible conformal predictors, each with a different $\alpha$ value, using the choice of calibration set by Straitouri et al. [19]. The results show that, while the mixture of MNLs tend to overestimates the average accuracy achieved by the predictions made by real experts, the average accuracy follows the same qualitative trend.

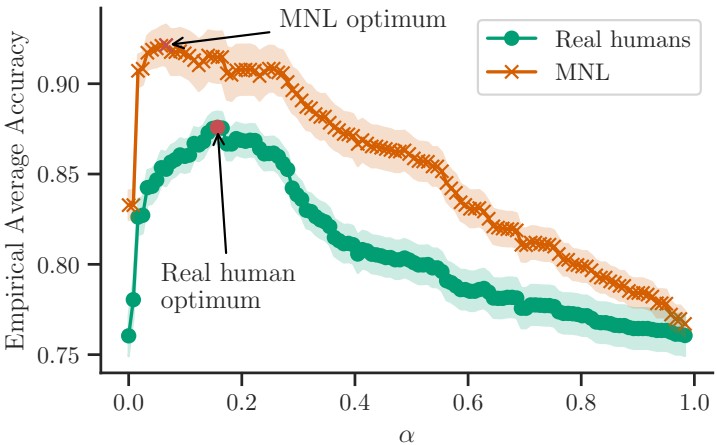

Figure 5: Average accuracy achieved by a simulated expert following the mixture of MNLs and by real human experts using the prediction sets constructed with all possible conformal predictors, each with a different $\alpha$ value, using the choice of calibration set by Straitouri et al. [19]. We highlight in red the highest average accuracy for both the simulated and the real humans.

# H Conformal Prediction under different $\alpha$ values

In this section, we estimate the average accuracy achieved by a (simulated) expert using the prediction sets constructed with conformal prediction (NAIVE, APS, RAPS and SAPS) on the ImageNet16H dataset under different $\alpha$ values. Figure 6 summarizes the results.

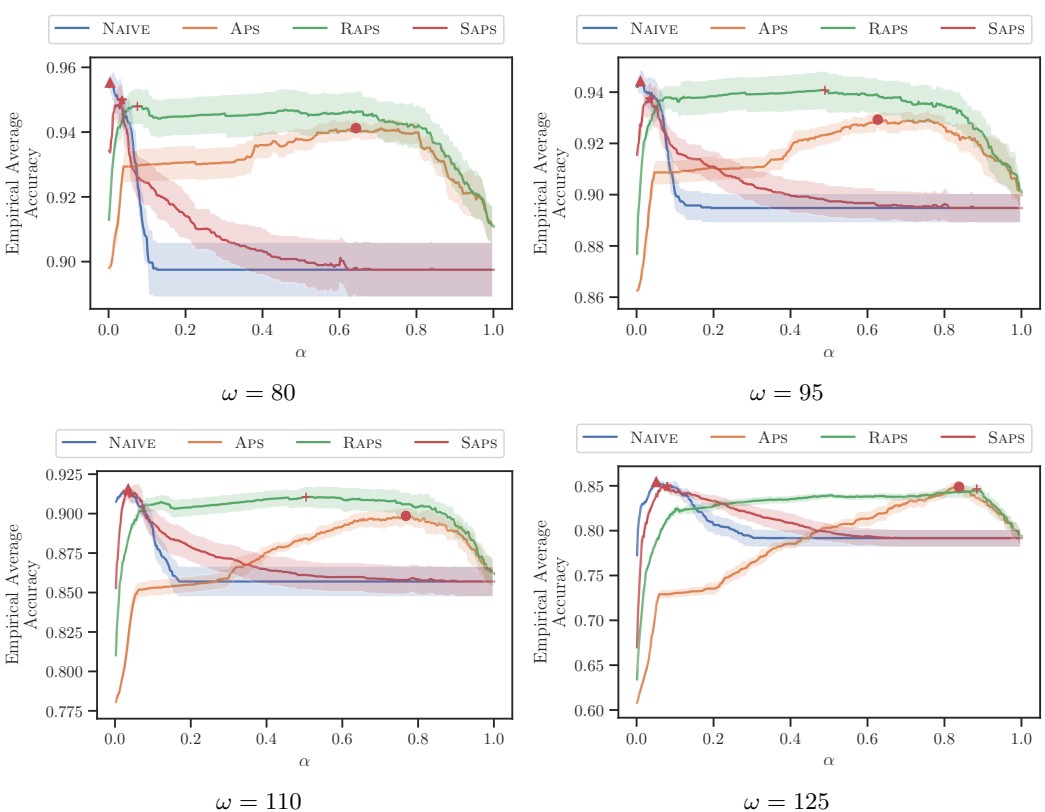

Figure 6: Average accuracy achieved by a (simulated) expert using the prediction sets constructed with conformal prediction (NAIVE, APS, RAPS and SAPS) on the ImageNet16H dataset under different $\alpha$ values. Each panel shows the average and standard error over 10 runs. We highlight with a red marker the highest average accuracy for the simulated humans under each conformal predictor.

