# OpenReview forum: "Towards Human-AI Complementarity with Prediction Sets"
_NeurIPS.cc/2024/Conference — NeurIPS 2024 poster_

### Official Review · Reviewer_trEi · 2024-06-27

**Soundness:** 3
**Presentation:** 3
**Contribution:** 3
**Rating:** 7
**Confidence:** 3

**Summary:**

The paper analyzes decision support systems based on prediction set algorithms. The authors show that: (i) the usage of conformal prediction techniques is generally sub-optimal in terms of accuracy; (ii) the problem of finding the optimal prediction sets under human assistance is NP-hard. Moreover, they provide (iii) a greedy algorithm that is guaranteed to find prediction sets that are better than those provided by conformal predictors. Experimental evaluation on synthetic and real data show the effectiveness of the considered approach.

**Strengths:**

The main strengths of the paper are:

1. the actual paper contribution is well framed;
2. the theoretical analysis is sound;
3. the proposed algorithm improves over existing approaches.

**Weaknesses:**

I think this work is a good paper, without major weaknesses, as it provides solid theoretical insights.
The concerns I have are mainly due to typos/details missing. I will point out here these and a few remarks that might be considered for the final version of the paper.

1. It seems to me that Table 2 and Figure 3 are missing the BRUTE FORCE baseline.
2. regarding the style of the paper, I found lines 135-146 very dense. Maybe providing a more concrete example (e.g., what could 1,2,3 represent?) might help the reader getting through it.
3. In Algorihm 1, I think adding a comment to the pseudo-code (from lines 4 to 13) could be useful
4. regarding the limitation section (evaluation) a useful reference might be [Stutz et al., 2023], where the authors evaluate the possibility that human experts might not be approximating the true probability distribution
5. the experimental analysis (on real data) could be enriched with other popular Learning-to-Defer datasets, such as Cifar10H or hatespeech.

[Stutz et al., 2023] - Stutz, David, Abhijit Guha Roy, Tatiana Matejovicova, Patricia Strachan, Ali Taylan Cemgil, and Arnaud Doucet. "Conformal prediction under ambiguous ground truth." Transactions on Machine Learning Research (2023).

**Questions:**

I have a couple of questions/remarks:

1. Can you elaborate a bit more on lines 91-93? I am not fully sure I understand the point there.
2. Can you add the results for BRUTE FORCE SEARCH in Table 2 and Figure 3?

**Limitations:**

The paper adequately discussed the limitations and societal impact.

---

> ### Author Rebuttal · Authors · 2024-08-02
>
> **[Lines 125-146 & Algorithm 1]** To ease readability, we will rewrite 135-146 and we will add comments to the pseudo-code in Algorithm 1.
>
> **[Limitation section]** In the limitation section under "Evaluation", we will add a discussion of and citation to Stutz et al., 2023.
>
> **[Other popular datasets]** We agree that it would be interesting to evaluate our greedy algorithm in other datasets besides ImageNet-16H. However, there is a scarcity of publicly available datasets with multiple expert predictions per sample, a relatively large number of samples, and more than two/three classes. In the suggested datasets, either the performance of the human experts on their own is very high (CIFAR-10H) or the number of classes is very small (hatespeech) for decision support systems based on prediction sets to be practical.
>
> **[Lines 91-93]** If we allow the expert to predict label values from outside the prediction set, the expert would have to decide when to believe that the true label is in the prediction set and thus predict a label from the prediction set, and when to question that the true label is in the prediction set and thus predict a label from outside the prediction set.
>
> In this context, it is worth pointing out that, as shown in a large-scale human subject study by Straitouri et al. (ICML 2024) [19], if we allow the expert to predict label values from outside the prediction sets, the number of predictions in which the prediction sets do not contain the true label and the experts succeed is consistently smaller than the number of predictions in which the prediction sets contain the true label and the experts fail, as shown in Figure 10 in their paper.
>
> **[Brute force search]** Since the greedy algorithm achieves the same performance as brute force search, we omitted brute force search from Table 1, Table 2 and Figure 3. However, we missed to clarify that in the caption of Table 2 and Figure 3. We will clarify that in the revised version of the paper.

---

> > ### Comment · Reviewer_trEi · 2024-08-08
> > **Response to the Rebuttal**
> >
> > I thank the authors for their clarifications. I have no further questions.
> > After reading the rebuttals and the other reviews, I am keeping my score as I think this is a sound paper.

---

### Official Review · Reviewer_QrrM · 2024-07-07

**Soundness:** 2
**Presentation:** 2
**Contribution:** 2
**Rating:** 5
**Confidence:** 3

**Summary:**

The authors first show that conformal prediction sets may not lead to human decision optimality.  The authors then introduce a greedy algorithm to generate candidate prediction sets that improve human decisions regarding the accuracy metric.

**Strengths:**

The authors find the sub-optimality of conformal prediction sets on providing candidates for human decisions. Thereby, they propose a novel method to produce prediction sets that helps to improve human prediction.

**Weaknesses:**

* The presentation somewhere is unclear:
  * Line 86. Please break the sentence properly.
  * Line 40/43/48: It is unclear for readers when the authors mention “optimal” multiple times but delay its explicit definition later.
  * Line 197: It is confusing when the authors refer to the role of $a$. What is the value of $a$?

* The authors claim they propose an efficient algorithm. However, I am not sure which part is efficient. Are there any numerical metrics, e.g., running time, supporting this contribution? Additionally, how should we understand this restriction of “for a large class of non-conformity scores and expert models” in line 51?

* Line 90:  But you also miss the possibility outside the prediction set, especially when the prediction set is not that good. I think the authors need to discuss the exploitation-exploration dilemma.

* The authors use the scores related to softmax and APS. Other papers propose alternative scores like RAPS and SAPS. I think they should be included.

**Questions:**

* Typo in the title: Predictions --> Prediction

* Did you include the results from the conformal prediction sets by varying the value of $\alpha$?

*  Why choose those values of $\omega$? I think their magnitudes are close. The authors may consider even smaller and larger values of $\omega$ to show its sensitivity.

**Limitations:**

Please see the above sections.

---

> ### Author Rebuttal · Authors · 2024-08-02
>
> **[Optimal]** We will define what we mean by optimal the first time we mention optimal in the revised version of the paper.
>
> **[Role of $a$]** In the generative model we used in the synthetic experiments, out of 20 features per sample, $d=4$ of these features correlate with the label value and thus are informative and the rest are just noise. In this context, "a" denotes the value of one of these four informative features. We will bring this information from Appendix C to the main in the revised version of the paper.
>
> **[Efficient algorithm]** The computational complexity of our greedy algorithm is polynomial on the size of the label set $|\mathcal{Y}|$, as shown in lines 171-176 in Section 4 and Appendix B. We will clarify that, by efficient, we refer to "computational efficiency" in the revised version of the paper.
>
> **[Class of non-conformity scores and expert models]** The guarantees of our greedy algorithm with respect to conformal prediction (Proposition 1) apply to any non-conformity score that is nonincreasing with respect to the classifier scores $f_y(x)$ and expert models parameterized by a mixture of multinomial logit models (MNLs). We will clarify this in the revised version of the paper.
>
> **[Exploitation-exploration dilemma]** Straitouri et al. (ICML 2024) [19] conducted a large-scale human subject study where they compared a setting where users are not allowed to select label values outside the prediction sets against another setting where users are allowed to select label values outside the prediction sets. They found that, in the latter setting, the number of predictions in which the prediction sets do not contain the true label and the experts succeed is consistently smaller than the number of predictions in which the prediction sets contain the true label and the experts fail, as shown in Figure 10 in their paper. As a consequence, humans perform better in the former setting than in the latter setting, as shown in Figure 3 in their paper. Following the reviewer's suggestion, we will include such a discussion in the revised version of the paper.
>
> **[RAPS and SAPS]** Following the reviewer's suggestion, we conducted extensive additional experiments using the best conformal predictors created with RAPS and SAPS [*], and we will include them as baselines in the revised version of the paper. Both methods present additional hyperparameters ($k_{reg}$ and $\lambda_{raps}$, for RAPS, and $\lambda_{saps}$ for SAPS) we optimize using a held-out validation set and the procedure outlined in the original papers (e.g., see Appendix E [49]).
>
> The empirical evaluation shows that their performance is worse than our greedy algorithm for synthetic and real-world tasks (Tables 1 and 2 of the rebuttal PDF). Moreover, RAPS and SAPS offer comparable performances to NAIVE and APS in all the tasks. The complete results can be found in the PDF attached to the general rebuttal comment above.
>
> [*] Huang, Jianguo, et al. "Conformal prediction for deep classifier via label ranking." ICML (2024).
>
> **[$\alpha$ value]** In Table 1 and Table 2, we report the results for the value of achieving the highest empirical success probability for each non-conformity score and classification task, as noted in lines 211-212 and 262-263. In Appendix G, for the experiments with real data, we report the result for all values of $\alpha$.
>
> **[$\omega$ values]** The ImageNet-16H dataset contains only images with the $\omega$ values we used and thus we cannot experiment with smaller or larger values.  However, we would like to point out that, even if the levels of phase noise $\omega$ seem close, the accuracy of the predictions made by humans on their own varies significantly among $\omega$ values, as shown in Figure 4S of Steyvers, Mark, et al. "Bayesian modeling of human–AI complementarity." Proceedings of the National Academy of Sciences 119.11 (2022): e2111547119.

---

> > ### Comment · Reviewer_QrrM · 2024-08-12
> >
> > I acknowledge the authors' response and I keep my initial score.

---

### Official Review · Reviewer_Dxna · 2024-07-07

**Soundness:** 3
**Presentation:** 2
**Contribution:** 3
**Rating:** 6
**Confidence:** 4

**Summary:**

The paper shows the conformal prediction set may not be the optimal set recommendation to humans if humans follow certain choice models. The authors then propose a greedy algorithm by modeling $P(y|x)$ and the choice model of humans assuming it follows MNL model. Authors compare the proposed method against the standard conformal prediction set under synthetic human experts and the proposed method has a slightly better performance compared to traditional conformal sets.

**Strengths:**

The authors consider conformal prediction in the human-in-the-loop setting, which is an important problem. The first part of the paper shows the conformal prediction set may not be the best recommendation set for humans, which is easy to understand since most conformal sets arrange the set in a ranked order and we can play with the human choice models to create an example that conformal sets may not be the best recommendation set.

**Weaknesses:**

The problem setting is not realistic: The authors do not allow humans to select outside the conformal prediction set. However, in the setups of most empirical successes of human-AI collaboration with conformal prediction, this is allowed. Similarly, if the authors do not allow humans to select outside the conformal prediction set, humans' value is greatly reduced and the optimal thing to do may be just to use fully automated AI prediction and in all the toy examples the authors provided, kicking humans out of the loop is the optimal system (humans only make things worse).

The theoretical analysis seems useless: I think the theoretical analysis is useless for two reasons: 1) while identifying the optimal set is NP-hard, in practice the metric we care about is $\mathbb{E} g(S|x)$, not identifying the optimal set. If an algorithm can get a good rate of convergence for this regret, then this problem is not hopeless, so I think authors need to show for all conformal prediction algorithms, what is the regret lower bound for $\mathbb{E} g(S|x)$; 2) while I can see that sometimes the label set can be large. In practice, the theoretical results may not be a big issue for many problems since most problems have small label set (binary or three classes). This negative results may not seem that severe as the authors presented in the paper.

The solution is disconnected and not useful in human-AI collaboration: 1) The proposed solution does not enjoy the distributionally-free guarantee, which is the main reason why people use conformal prediction. I would expect authors to provide a conformal prediction algorithm that is human-centered, rather than directly switch lanes to traditional prediction methods. 2) The proposed solution requires $P(y|x)$ and the true human choice model, which is too strong to be realistic. If I know $P(y|x)$, why should I involve humans in the loop anymore (recall that authors can restrict humans only select from prediction set so humans are not necessary in the system). The optimal strategy would be directly use $P(y|x)$ to select actions.

Baselines: For human-AI collaboration tasks, I expect to see the proposed solution is better than human working alone or AI working alone. The authors should compare with AI only baseline using $P(y|x)$. Based on the toy example and my current understanding of the paper, the proposed solution cannot beat AI only baseline.

**Questions:**

See weakness.

**Limitations:**

Yes

---

> ### Author Rebuttal · Authors · 2024-08-02
>
> **[Problem setting]** Straitouri et al. (ICML 2024) [19] conducted a large-scale human subject study where they compared the setting we adopted, where users are not allowed to select label values outside the conformal prediction sets, against the setting the reviewer suggests, where users are allowed to select label values outside the sets. The results of their study (refer to Figure 3 in their paper) suggest that humans perform better under the setting we adopted, which Straitouri et al. refer to as the strict implementation of their system, than under the setting the reviewer suggests, which they refer to as the lenient implementation of their system.
>
> Further, we would like to clarify that, in our experiments, we found that kicking humans out of the loop is **not** the optimal system (humans do **not** make things worse) in the setting we adopted. More specifically, in the experiments with synthetic data, both the accuracy of the classifier $P(Y'=Y)$ we used (shown in the top row of Table 1) and the accuracy of the human (rows under NONE in Table 1) are always lower than the accuracy of the human using the prediction sets provided by our greedy algorithm (shown in rows under GREEDY of Table 1). This suggests that even if the human has low performance overall, it may still be beneficial to use prediction sets tailored to the performance of humans rather than "kicking them out".
>
> In the experiments with real data, the accuracy of the (fine-tuned) VGG-19 classifier we used ($0.896$ for $w=80$, $0.894$ for $\omega=95$, $0.857$ for $\omega=110$ and $0.792$ for $\omega=125$) and the accuracy of the human working alone ($0.9$ for $\omega=80$, $0.859$ for $\omega=95$, $0.771$ for $\omega=110$ and $0.603$ for $\omega=125$), which we missed reporting in our paper, are also always lower than the accuracy of the humans using the prediction sets provided by our greedy algorithm (shown in rows under GREEDY in Table 2). In the revised version of the paper, we will report the accuracy of the (fine-tuned) VGG-19 classifier we used and of the human working alone.
>
> **[Theoretical analysis]** Our hardness analysis implies that finding the optimal prediction sets that maximize the metric $\mathbb{E}[g(\mathcal{S}|x)]$ is NP-hard, and even finding a prediction set that approximates this to a reasonable factor (e.g. a constant factor) is also NP-hard. As a consequence, there is no polynomial-time algorithm with a sublinear regret (i.e., a _good_ regret) with respect to the (oracle) algorithm that creates optimal prediction sets that maximize $\mathbb{E}[g(\mathcal{S}|x)]$. We will clarify this in the revised version of the paper.
>
> While there are certainly many problems with small label sets, there are also many problems with large label sets. In fact, two of the most popular benchmark datasets in the machine learning literature used by thousands of papers, ImageNet and its more commonly known subset ImageNet-1k, contain over 20,000 and 1,000 different label values, respectively. As an additional example of tasks with large label space, we mention a clinical text annotation task where each span of text has to be mapped to a _"concept label in a large (>400,000) medical vocabulary"_ [1].
>
> [1] Levy, Ariel, et al. "Assessing the impact of automated suggestions on decision making: Domain experts mediate model errors but take less initiative." Proceedings of the 2021 CHI Conference on Human Factors in Computing Systems. 2021.
>
> **[Distributionally-free guarantees of conformal prediction]** One of the main contributions of our paper is to demonstrate that, in human-AI collaboration, the distributionally-free guarantees offered by conformal prediction may be insufficient to achieve optimal performance. More specifically, in Section 3, we show that, under common choices of non-conformity scores, there are many data distributions for which the optimal prediction set under which the human expert achieves the highest accuracy **cannot** be constructed by **any** conformal predictor. As suggested by the reviewer, one may think of developing human-centered conformal predictors that incorporate information about the distribution of experts’ predictions in the definition of the non-conformity score. However, as argued in lines 118-122, our hardness results show that one cannot expect to fully close the performance gap with such human-centered conformal predictors.
>
> **[Knowledge of $P(Y|X)$]** Even if one knows the conditional distribution of the ground-truth label $P(Y|X)$, one may benefit from involving humans in the loop if the humans have access to additional features besides X, as pointed out in footnote 4. For instance, in the example in lines 106-117, the optimal prediction set under which the human achieves the highest accuracy does not contain the label value with the highest $P(Y|X)$ value. Further, under this prediction set, the human achieves higher accuracy than a classifier that picks the label value with the highest $P(Y|X)$ value (0.6 vs 0.4).
>
> **[Human-only and AI-only baselines]** In our experiments, both with synthetic and real data, our proposed solution is better than humans working alone or AI working alone. Regarding the comparison with AI working alone, please, refer to our previous reply under [Problem setting]. Regarding the comparison with human working alone, in the experiments with synthetic data, the accuracy of the human working alone is reported under "NONE" and, in the experiments with real data, the accuracy of the human working alone is $0.9$ for $\omega=80$, $0.859$ for $\omega=95$, $0.771$ for $\omega=110$ and $0.603$ for $\omega=125$, which we missed reporting in our paper and we will report in the revised version of the paper.

---

> > ### Comment · Reviewer_Dxna · 2024-08-08
> >
> > Thank authors for the clarification. I revisited the toy example and realize the human+AI is actually better than the ground-truth set ranked by probability, this is an interesting example, which addressed my concern. I raised my score accordingly.

---

### Official Review · Reviewer_9FF1 · 2024-07-12

**Soundness:** 2
**Presentation:** 3
**Contribution:** 3
**Rating:** 6
**Confidence:** 4

**Summary:**

This paper aims to construct optimal prediction sets under which experts can achieve the highest accuracy. The authors claim that human experts cannot attain maximum accuracy with the prediction sets generated by conformal predictors. To address this issue, the paper proposes an efficient greedy algorithm based on maximum marginal gain to find prediction sets that outperform those generated by conformal predictors. The paper offers two main theoretical contributions: the first proves that finding the optimal prediction set is an NP-hard problem, while the second demonstrates that the proposed method enables experts to achieve higher accuracy than conformal predictors. Empirical results further validate the effectiveness of the proposed approach.

**Strengths:**

1. The paper is well-motivated and easy to follow.

2. The authors provide a theoretical analysis for their motivation and offer a theoretical guarantee for the superior performance of the proposed greedy algorithm.

3. The paper presents an extensive set of experiments, including both synthetic and real data.

**Weaknesses:**

1. Further validation on more realistic datasets, such as ImageNet and CIFAR100, could strengthen the main points of the paper.

2. The experiments lack comparison with other classical score functions, such as Regularized Adaptive Prediction Sets.

**Questions:**

1. In Figure 3, how is the Empirical Success Probability for each image calculated?

2. In line 210, why does the score function of APS discard the random variable? In other words, does the random variable affect the performance of the empirical average test accuracy?

3. Can you report the empirical coverage of the Greedy algorithm, since valid coverage is the fundamental guarantee for conformal prediction?

**Limitations:**

They are adequately discussed.

---

> ### Author Rebuttal · Authors · 2024-08-02
>
> **[More realistic datasets]** The dataset ImageNet-16H is among the only publicly available datasets that we found containing multiple expert predictions per sample, a relatively large number of samples, more than two/three classes and a reasonable level of difficulty. The suggested datasets, ImageNet and CIFAR100, do not contain multiple expert predictions per sample. Another dataset for a multiclass classification task is CIFAR-10H which also has multiple expert predictions per sample and 10 classes. However, we found that the experts on their own already achieve high accuracy ($\sim 0.95$), making it a less challenging scenario than ImageNet-16H.
>
> **[RAPS]** Following the reviewer's suggestion, we conducted extensive additional experiments using the best conformal predictors created with RAPS and SAPS [*], and we will include them as baselines in the revised version of the paper. Both methods present additional hyperparameters ($k_{reg}$ and $\lambda_{raps}$, for RAPS, and $\lambda_{saps}$ for SAPS), which we optimized using a held-out validation set and the procedure outlined in the original papers (e.g., see Appendix E [49]).
>
> The empirical evaluation shows that their performance is worse than our greedy algorithm for synthetic and real-world tasks (Tables 1 and 2 of the rebuttal PDF). Moreover, RAPS and SAPS offer comparable performances to NAIVE and APS in all the tasks. The complete results can be found in the PDF attached to the general rebuttal comment above.
>
> [*] Huang, Jianguo, et al. "Conformal prediction for deep classifier via label ranking." ICML (2024).
>
>
> **[Empirical success probability]** For each image $x$ and prediction set $\mathcal{S}$, we estimate the empirical success probability using the mixture of multinomial logit models (MNLs), i.e., $P_{\mathcal{S}}(\hat Y = y | X=x, Y=y) = \frac{C_{yy}}{\sum_{y' \in \mathcal{S}} C_{y'y}}$, where $\hat{Y}$ denotes the prediction by the human, $Y$ denotes the true label and, in the experiments with real data, the confusion matrix $\mathbf{C}$ is estimated using predictions made by real human experts on their own.
>
> **[Randomization in APS]** We did not find the randomization in APS, which is just needed to achieve $1-\alpha$ coverage exactly, to influence the empirical success probability in our experiments. Therefore, for simplicity, we decided to omit it. We will clarify this in the revised version of the paper.
>
> **[Empirical coverage of Greedy]** We will report the empirical coverage achieved by the greedy algorithm and the best conformal predictors used as baselines in the revised version of the paper. Here below we show the empirical coverage for the synthetic tasks.
>
> In summary, prediction sets constructed with our greedy algorithm present a comparable empirical coverage with respect to the best conformal predictors (Naive and APS) in both synthetic and ImageNet-16H tasks. Moreover, in those few settings in which the conformal predictors coverage is higher (e.g., $P(\hat{Y} = Y) = 0.3$ and $\gamma = 0.3$), the conformal predictors' empirical success probability is lower (see Table 1 of the main paper), which underlines how coverage alone can be a bad proxy to estimate the empirical human accuracy.  Moreover, please note how the empirical success probability of the human acting alone with the full label set, that achieves perfect coverage (1.0), is always worse than the combination of human + predictions sets in all of our experiments.
>
> | $\gamma$ | Method |   $\mathbb{P}[Y' = Y] = 0.3$  |   $\mathbb{P}[Y' = Y] = 0.5$  |   $\mathbb{P}[Y' = Y] = 0.7$  |   $\mathbb{P}[Y' = Y] = 0.9$  |
> |:-----:|--------|:-----------------------------:|:-----------------------------:|:-----------------------------:|:-----------------------------:|
> |  0.3  | Naive  | $0.637 \scriptstyle\pm 0.066$ | $0.802 \scriptstyle\pm 0.058$ | $0.908 \scriptstyle\pm 0.020$ | $0.973 \scriptstyle\pm 0.007$ |
> |       | Aps    | $0.603 \scriptstyle\pm 0.083$ | $0.804 \scriptstyle\pm 0.045$ | $0.900 \scriptstyle\pm 0.026$ | $0.967 \scriptstyle\pm 0.006$ |
> |       | Greedy | $0.502 \scriptstyle\pm 0.024$ | $0.764 \scriptstyle\pm 0.019$ | $0.920 \scriptstyle\pm 0.012$ | $0.976 \scriptstyle\pm 0.004$ |
> |  0.5  | Naive  | $0.583 \scriptstyle\pm 0.104$ | $0.770 \scriptstyle\pm 0.044$ | $0.897 \scriptstyle\pm 0.021$ | $0.968 \scriptstyle\pm 0.009$ |
> |       | Aps    | $0.557 \scriptstyle\pm 0.100$ | $0.741 \scriptstyle\pm 0.036$ | $0.879 \scriptstyle\pm 0.016$ | $0.961 \scriptstyle\pm 0.007$ |
> |       | Greedy | $0.489 \scriptstyle\pm 0.027$ | $0.732 \scriptstyle\pm 0.015$ | $0.902 \scriptstyle\pm 0.013$ | $0.970 \scriptstyle\pm 0.003$ |
> |  0.7  | Naive  | $0.535 \scriptstyle\pm 0.104$ | $0.676 \scriptstyle\pm 0.047$ | $0.823 \scriptstyle\pm 0.044$ | $0.938 \scriptstyle\pm 0.013$ |
> |       | Aps    | $0.519 \scriptstyle\pm 0.115$ | $0.661 \scriptstyle\pm 0.036$ | $0.853 \scriptstyle\pm 0.015$ | $0.941 \scriptstyle\pm 0.013$ |
> |       | Greedy | $0.473 \scriptstyle\pm 0.017$ | $0.696 \scriptstyle\pm 0.014$ | $0.861 \scriptstyle\pm 0.014$ | $0.958 \scriptstyle\pm 0.005$ |
> |  1.0  | Naive  | $0.499 \scriptstyle\pm 0.075$ | $0.608 \scriptstyle\pm 0.034$ | $0.750 \scriptstyle\pm 0.025$ | $0.905 \scriptstyle\pm 0.014$ |
> |       | Aps    | $0.453 \scriptstyle\pm 0.062$ | $0.631 \scriptstyle\pm 0.038$ | $0.806 \scriptstyle\pm 0.026$ | $0.912 \scriptstyle\pm 0.013$ |
> |       | Greedy | $0.457 \scriptstyle\pm 0.024$ | $0.664 \scriptstyle\pm 0.015$ | $0.839 \scriptstyle\pm 0.014$ | $0.951 \scriptstyle\pm 0.006$ |

---

> > ### Comment · Reviewer_9FF1 · 2024-08-11
> >
> > Thank you for your response. I still have only one concern:
> >
> > [**Empirical coverage of Greedy**] Regarding the selection of $\alpha$, is it justifiable to report results for the $\alpha$ value at which the expert attains the highest average test accuracy? The test set is intended for evaluating the performance of various methods, not for choosing hyper-parameters. Therefore, it may be more appropriate to adjust $\alpha$ based on a separate hold-out dataset.

---

> > > ### Author Response · Authors · 2024-08-12
> > >
> > > **[Selection of $\alpha$]** We would like to thank the reviewer for their follow-up message. In our experiments, we report the results for the $\alpha$ value at which the expert attains the highest average accuracy because our goal was to show that our greedy algorithm achieves better results than conformal prediction for _any_ value of $\alpha$. However, we agree with the reviewer that, in practice, one would need to select $\alpha$ using a held-out dataset. Therefore, to avoid any misunderstanding, we will add a clarification and, if the reviewer feels it is necessary, we will also add an Appendix where we adjust $\alpha$ based on a separate held-out set.

---

> > > > ### Comment · Reviewer_9FF1 · 2024-08-12
> > > >
> > > > Thank the authors for the clarification. I have no other concerns. Adding complementary experiments can make the experiment results more convincing. I maintain the rating of accept.

---

### Author Rebuttal · Authors · 2024-08-02

We would like to thank the reviewers for their careful and insightful comments, which will help improve our paper. Please, find a point-by-point response below and a one-page pdf with additional results attached.

---

### Decision · Program_Chairs · 2024-09-25

**Decision:**

Accept (poster)

**Comment:**

All four reviewers are in favor of acceptance. They appreciated the submission's main contributions: finding that conformal prediction yields suboptimal prediction sets in terms of human accuracy, showing the NP-hardness of finding optimal prediction sets, and showing that a proposed greedy algorithm is guaranteed to improve upon conformal prediction. The theoretical nature of the last two contributions was especially appreciated. Other strengths include the experiments involving real experts and the writing being easy to follow.

The rebuttal satisfactorily addressed several issues, including the value of restricting humans to selecting from the prediction set, experimenting on additional datasets, and a comparison with RAPS and SAPS. It also answered reviewer questions, clarified some misunderstandings, and promised to report additional results, for example regarding empirical coverage and selection of $\alpha$. I hope that the authors will incorporate these elements into the camera-ready version. I do note one reviewer comment about the necessity of knowing $P(Y|X)$ and the human choice model. It would be good to comment on this in the final version.